

# Cranial osteology of the Early Cretaceous turtle *Pleurosternon bullockii* (Paracryptodira: Pleurosternidae)

Serjoscha W. Evers, Yann Rollot and Walter G. Joyce

Department of Geosciences, University of Fribourg, Fribourg, Switzerland

## ABSTRACT

*Pleurosternon bullockii* is a turtle from the Early Cretaceous of Europe known from numerous postcranial remains. Only one skull has so far been referred to the species. *Pleurosternon bullockii* belongs to a group of turtles called pleurosternids, which is thought to include several poorly known taxa from the Late Jurassic and Early Cretaceous of Europe and North America. Pleurosternids and baenids, a group of North American turtles that lived from the Late Cretaceous to the Eocene, define a clade called Paracryptodira. Additionally, Paracryptodira likely includes compsemydids, and, potentially, helochelydrids. Character support for Paracryptodira is relatively weak, and many global phylogenetic studies fail to support paracryptodiran monophyly altogether. Proposed paracryptodiran synapomorphies are largely cranial, despite the poor characterization of pleurosternid cranial material. In addition to their questionable monophyly, the global position of paracryptodires is debated. Early studies suggest crown-turtle affinities, but most phylogenies find them as stem-turtles, irrespective of their monophyly. Here, we document the cranial osteology of *Pleurosternon bullockii* with the use of three-dimensional models derived from segmenting high-resolution X-ray micro-computed tomography (CT) scans. *Pleurosternon bullockii* has a primitive basipterygoid region of the skull, but a cryptodire-like acustico-jugular region. A surprising number of similarities with pleurodires exist, particularly in the laterally expanded external process of the pterygoid and in the posterior orbital wall. Our observations constitute an important step toward a phylogenetic re-evaluation of Paracryptodira.

## INTRODUCTION

Paracryptodires are often referred to as 'one of the three primary clades of (crown) turtles' (e.g., *Joyce, 2017*). This assessment is primarily based on unresolved phylogenetic positions between Pleurodira, Cryptodira, and Paracryptodira among early global phylogenetic studies (e.g., *Joyce, 2007*). However, most recent phylogenetic analyses recover paracryptodires as stem-turtles, just outside of the turtle crown (e.g., *Sterli, 2010*; *Rabi et al., 2014*; *Cadena & Parham, 2015*; *Zhou & Rabi, 2015*; *Joyce et al., 2016*; *Evers & Benson, 2019*). Paracryptodires were originally defined as a group of turtles that includes baenids, pleurosternids (including 'Glyptopsidae' of *Gaffney, 1972*), *Compsemys victa*,

Corresponding author
Serjoscha W. Evers,
serjoscha.evers@googlemail.com

and *Kallokibotion bajazidi* (*Gaffney, 1975a*). Although only one formal phylogenetic analysis has since found these exact turtles in a monophyletic group (a *Lyson & Joyce, 2011* analysis of the matrix of *Gaffney et al., 2007*), a sister relationship of baenids with pleurosternids has been found frequently (e.g., *Gaffney et al., 2007*; *Joyce, 2007*; *Lyson & Joyce, 2011*; *Pérez-García, Royo-Torres & Cobos, 2015*; *Joyce et al., 2016*; *Joyce & Rollot, 2020*). As a consequence, most authors associate the term 'Paracryptodira' with a clade that either exclusively includes baenids and pleurosternids (e.g., *Joyce, 2007*), or a clade that minimally includes these clades, variously joined by compsemydids, helochelydrids, or indeterminate paracryptodiran taxa (*Lyson & Joyce, 2011*). *Lyson & Joyce (2011)* formalized this notion of Paracryptodira by coining a stem-based phylogenetic definition that includes *Pleurosternon bullockii* and *Baena arenosa* as its internal specifiers. Character support for Paracryptodira is relatively weak. The main osteological feature that has historically been used to unite paracryptodires is a far anteriorly positioned foramen for the entry of the internal carotid artery (foramen posterius canalis carotici interni; *Rabi et al., 2013*) formed midway along the parabasisphenoid-pterygoid suture (*Gaffney, 1975a*). This feature persists to be cited as diagnostic for paracryptodires (see review of *Joyce & Anquetin, 2019*), although the foramen has repeatedly been interpreted to be absent altogether in at least some paracryptodires (*Sterli et al., 2010*; *Rabi et al., 2013*; *Anquetin & André, 2020*). Additionally, many of the other originally proposed characters in support of the group are symplesiomorphically present in Testudinata (e.g., presence of nasal, reduced prefrontal exposure on skull roof, presence of mesoplastra, presence of paired gular scutes; *Gaffney, 1975a*). Some proposed synapomorphies, such as the reduction of the palatine artery (*Gaffney, 1975a*) or a reduced fenestra perilymphatica (*Joyce, 2007*), are, similar to the aforementioned position of the foramen posterius canalis carotici interni midway along the parabasisphenoid-pterygoid suture (*Gaffney, 1975a*; *Joyce & Anquetin, 2019*), not universally present in paracryptodires (e.g., *Anquetin & André, 2020*; this study) and should thus be re-evaluated on a broader scale. Ambiguity in character support of Paracryptodira is also implicit from many global phylogenetic studies that fail to retrieve the constituent taxa as monophyletic altogether (*Sterli, 2010*; *Anquetin, 2012*; *Sterli, Pol & Laurin, 2013*; *Cadena & Parham, 2015*; *Rabi et al., 2014*; *Zhou & Rabi, 2015*; *Joyce et al., 2016*; *Evers & Benson, 2019*). Although most frequently cited paracryptodiran synapomorphies are cranial characters, non-baenid paracryptodiran skulls are poorly known (*Joyce & Anquetin, 2019*). Five taxa have relatively complete skulls, including the compsemydid *Compsemys victa* (*Lyson & Joyce, 2011*), the indeterminate paracryptodire *Uluops uluops* (*Carpenter & Bakker, 1990*), and the pleurosternids *Glyptops ornatus* (*Gaffney, 1979a*), *Dorsetochelys typocardium* (*Evans & Kemp, 1976*) and *Pleurosternon bullockii* (*Evans & Kemp, 1975*). Here, we re-examine UMZC T1041, the cranium of *Pleurosternon bullockii* that was originally described as *Mesochelys durlstonensis* by *Evans & Kemp (1975)*, based on high-resolution micro-computed tomography (CT) scanning. We describe the cranium in detail and compare it to available material of other paracryptodires, and, where relevant, other groups of turtles. Thorough anatomical characterizations like the one presented here form the base work for phylogenetic assessments. Our description and comparisons provide anatomical evidence for unique morphologies of pleurosternids, which likely include at

least *Uluops uluops* besides *Pleurosternon bullockii* and *Glyptops ornatus*. Pleurosternids show a mixture of plesiomorphic and derived features, tentatively supporting their inferred position as stem-taxa that are relatively close to the crown-node of turtles. We hope that our descriptions lead to the coding of novel phylogenetic characters, that further scrutinize the monophyly and global position of paracryptodires.

## MATERIAL & METHODS

High-resolution X-ray computed tomography (CT) scans were obtained for the cranium of UMZC T1041, which is the holotype of *Mesochelys durlstonensis* (*Evans & Kemp, 1975*). UMZC T1041 was found in the Early Cretaceous (Berriasian) Purbeck Limestone Group in Durlston Bay, United Kingdom (*Evans & Kemp, 1975*). *Gaffney & Meylan (1988)* referred the material to *Pleurosternon bullockii Owen, 1842*, albeit without justification. *Milner (2004)* compared the shell remains associated with the cranium of *Mesochelys durlstonensis* with those of *Pleurosternon bullockii*, and presented anatomical evidence for the synonymy suggested by *Gaffney & Meylan (1988)*. This has generally been accepted ever since (see review of *Joyce & Anquetin (2019)*). Scans of UMZC T1041 were obtained by Roger Benson in 2017 at the Cambridge Biotomography Center, using a X-Tek H 225 μCT scanner (Nikon Metrology, Tring, UK). The cranium was scanned using a beam energy of 130 kV, a current of 250 μA, 500 ms exposure time, 1 frame per 1400 projections, and no filter, resulting in a voxel size of 0.03315 mm. The resulting CT-scans were segmented in the software Mimics (v. 16.0–19.0; http://biomedical.materialise.com/mimics), and 3D models were exported as .ply files. Figures of digital renderings were compiled using the software Blender v. 2.71 (www.blender.org/) . CT-slice data as well as the 3D models are deposited at MorphoSource (*Evers, 2020*).

We mostly compare UMZC T1041 to a selection of likely paracryptodires from the Late Jurassic to Cretaceous of North America and Europe based on description published in the literature, in particular *Arundelemys dardeni* (as described by *Lipka et al., 2006*), *Compsemys victa* (as described by *Lyson & Joyce, 2011*), *Dorsetochelys typocardium* (as described by *Evans & Kemp, 1976* under the name *Dorsetochelys delairi*), *Eubaena cephalica* (as described by *Rollot, Lyson & Joyce, 2018*), *Glyptops ornatus* (as described by *Gaffney, 1979a* under the name *Glyptops plicatulus*), and *Uluops uluops* (as described by *Carpenter & Bakker, 1990*). For comparisons with *Dorsetochelys typocardium*, we used a 3D model of the holotype (DORCM G.00023) that was made available under a CC-BY-NC-SA license by the "GB3D Type Fossils" projects hosted by the British Geological Survey at http://www.3d-fossils.ac.uk. The comparisons with *Compsemys victa* (based on UCM 53971) and *Uluops uluops* (based on UCM 49223) were complemented by CT scans of these specimens. The original descriptions are cited for all previously described features, but the specimen numbers are cited when novel observations regarding these taxa are made based on these scans. These scans will be further discussed and made public in a forthcoming paper.

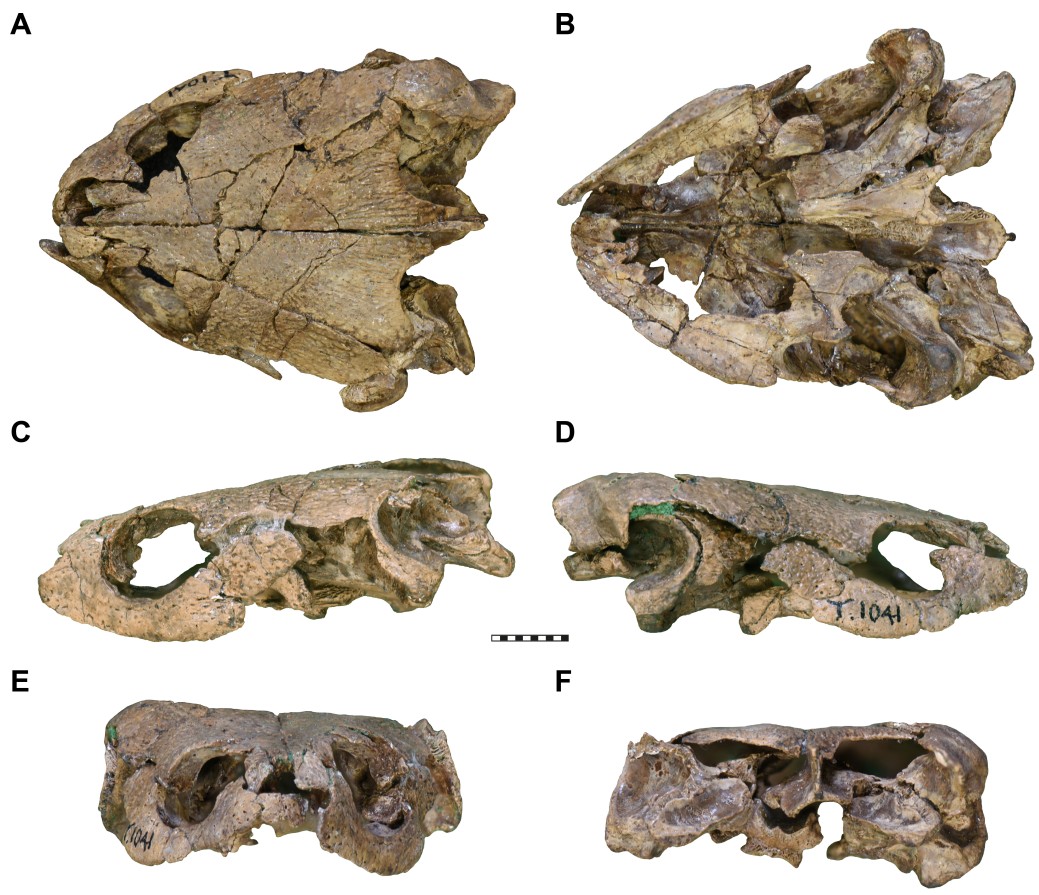

**Figure 1** **Photographs of UMZC T1041, cranium of *Pleurosternon bullockii*.** (A) dorsal view. (B) Ventral view. (C) Left lateral view. (D) Right lateral view. (E) Anterior view. (F) posterior view. Scale bar equals 10 mm.

## DESCRIPTION

**General comments.** The cranium of UMZC T1041 disarticulated into several blocks of fully to partially articulated bones, which were then displaced to give the skull a crushed appearance (Figs. 1 and 2). The 3D anatomy of individual bones is nevertheless near perfect. For instance, the left pterygoid, prootic, opisthotic, quadrate, and the parabasisphenoid form a block in which these bones are still mostly preserved in their original positions relative to one another. The parabasisphenoid, however, is shifted medially relative to the pterygoid.

The skull of UMZC T1041 is relatively low dorsoventrally, relatively narrow mediolaterally, but elongate anteroposteriorly (Figs. 1 and 2). The orbits face more laterally than in *Glyptops ornatus* (*Gaffney, 1979a*), but less so than in *Compsemys victa* (*Lyson & Joyce, 2011*). The entire skull roof and lateral side of the skull of UMZC T1041 is covered by the dense surface sculpturing that is characteristic of many paracryptodires (e.g., *Pérez-García, 2014*; *Joyce & Lyson, 2015*; *Joyce & Anquetin, 2019*). Although superficially the surface sculpturing of paracryptodires is very similar among species, some differences

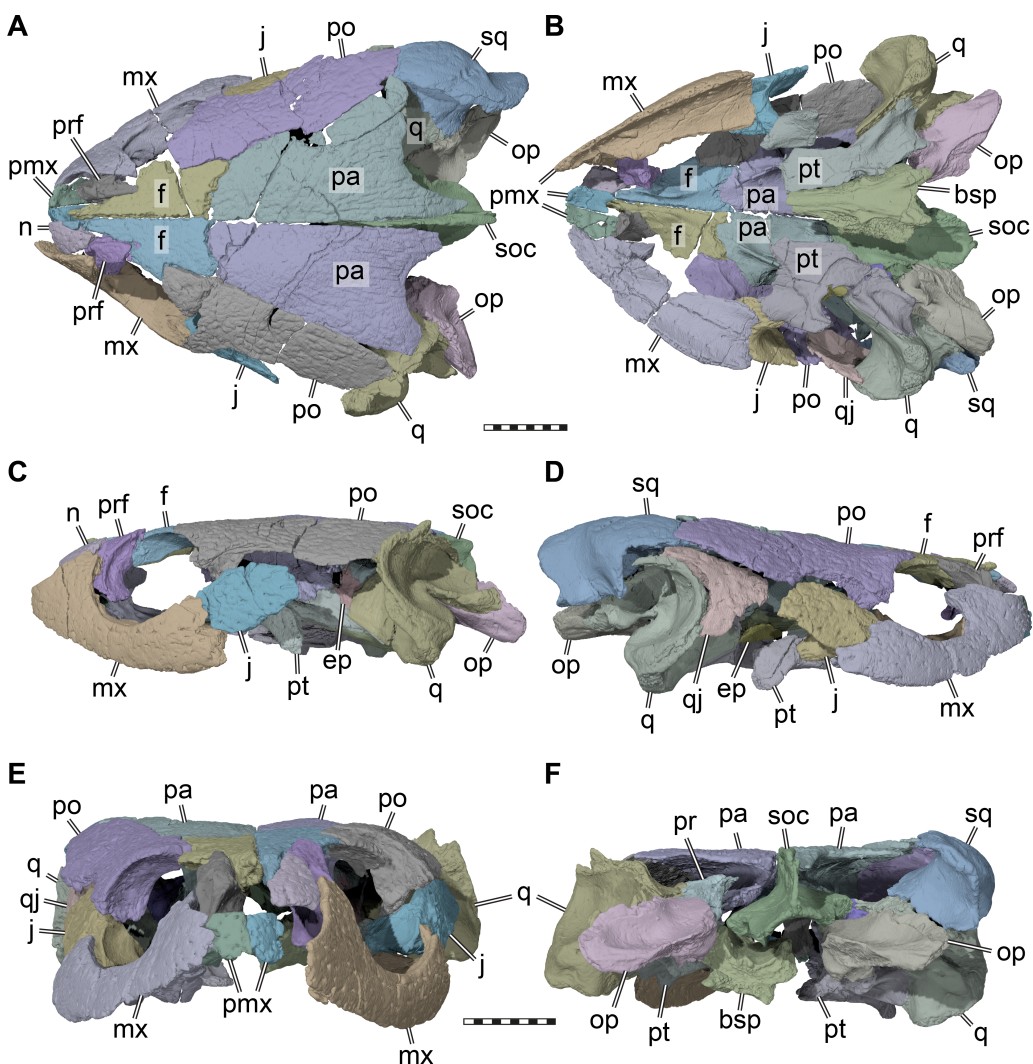

**Figure 2** **Three dimensional renderings of the cranium of *Pleurosternon bullockii* (UMZC T1041).** (A) Dorsal view. (B) Ventral view. (C) Left lateral view. (D) Right lateral view. (E) Anterior view. (F) Posterior view. Abbreviations: bsp, parabasisphenoid; ep, epipterygoid; f, frontal; j, jugal; mx, maxilla; n, nasal; op, opisthotic; pa, parietal; pmx, premaxilla; po, postorbital; pr, prootic; prf, prefrontal; pt, pterygoid; q, quadrate; qj, quadratojugal; soc, supraoccipital; sq, squamosal. Scale bars equal 10 mm.

can be made out. For instance, although the main textural element in both *Uluops uluops* (*Carpenter & Bakker, 1990*; UCM 53971) and *Pleurosternon bullockii* (UMZC T1041) are small and irregular tubercles, these transition into a ridge-like arrangement on the posterior margin of the parietal in UMZC T1041 (Fig. 2A), but not in *Uluops uluops* (UCM 53971). The skull emarginations of UMZC T1041 are moderately developed: the cheek emargination is relatively deep, reaching about mid-orbit in lateral view. The jugal and quadratojugal margins are strongly concave and the maxilla and quadrate are included in the emargination (Figs. 2C–2D). The same morphology is seen in *Uluops uluops* (UCM 53971). The medial portion of the upper temporal emargination of UMZC T1041 is visible

as a concavity developed along the posterior margin of the parietal that exposes only a small part of the floor of the adductor chamber (Fig. 2A), again as in *Uluops uluops* (UCM 53971). The lateral portions of the upper temporal emargination that were likely formed by the squamosal are damaged. Squamosal and supraoccipital crests are comparatively strongly developed, providing elongate attachment sites for the adductor musculature.

**Nasal.** Only the left nasal of UMZC T1041 is preserved (Figs. 2A, 2C, 2E). However, as the left nasal is in articulation with the frontal, prefrontal, and maxilla on the left side, the anterior region of the skull roof can be completely reconstructed. The nasal is a triangular element that tapers posteriorly and inserts into a gap between the prefrontal and frontal. The long anterior frontal processes separate the paired nasals posteriorly, but anteriorly a short nasal-nasal suture can be inferred, which is generally present in basal paracryptodires (*Gaffney, 1972*; *Joyce & Lyson, 2015*; *Joyce & Anquetin, 2019*).

**Prefrontal.** Both prefrontals of UMZC T1041 are preserved (Fig. 2). The prefrontal has a small skull roof exposure in UMZC T1041, and is restricted to a small area at the anterodorsal orbital margin, contacting the nasal, frontal, and maxilla. *Evans & Kemp (1975)* also reported a contact with the vomer. The relevant part of the specimen has since been lost, but the side of the descending process makes this contact highly plausible. Medially, the frontals hinder both prefrontals from a mutual contact. Anterolaterally, the prefrontal has a deep vertical groove for the reception of the ascending process of the maxilla, which is partially exposed on both sides due to partial disarticulation. However, on the left side, the maxilla and prefrontal are articulated enough to see that a small foramen orbito-nasale was laterally and dorsally framed by both bones. As the vomer and palatine bones are absent, the medial margin of the foramen is unknown. However, the foramen was medially probably closed by the palatine, as in *Uluops uluops* (UCM 53971).

**Frontal.** Both frontals are preserved (Fig. 2). The frontal contacts the nasal, prefrontal, postorbital, and parietal. The frontal is a relatively elongate, but narrow bone in UMZC T1041: the anterior portion forms long and narrow processes that divide the internasal contact for half of the nasal length (Fig. 2A). A process that partially separates the nasals is only slightly present in *Dorsetochelys typocardium* (*Evans & Kemp, 1976*) and *Arundelemys dardeni* (*Lipka et al., 2006*), but otherwise generally absent in other paracryptodires, such as *Glyptops ornatus* (*Gaffney, 1979a*), *Compsemys victa* (*Lyson & Joyce, 2011*), or baenids (*Gaffney, 1972*; *Joyce & Lyson, 2015*). The frontal contributes to the orbit in UMZC T1041 by means of a broad lateral process (Fig. 2A). Ventrally, both frontals form a narrow sulcus olfactorius that extends anteriorly into the fossa nasalis (Fig. 3). The right and left crista cranii diverge posteriorly from the midline and become shallower near the suture with the parietal.

**Parietal.** Both parietals are preserved in UMZC T1041 (Fig. 2). The parietal forms the posterior and central parts of the skull roof and contacts the frontal anteriorly, the postorbital laterally, and almost certainly the squamosal posterolaterally. Ventrally, the parietal contacts the supraoccipital, prootic, epipterygoid, and possibly the pterygoid (Fig. 4B). The parietal forms a tapering posteromedian process, which seems to have

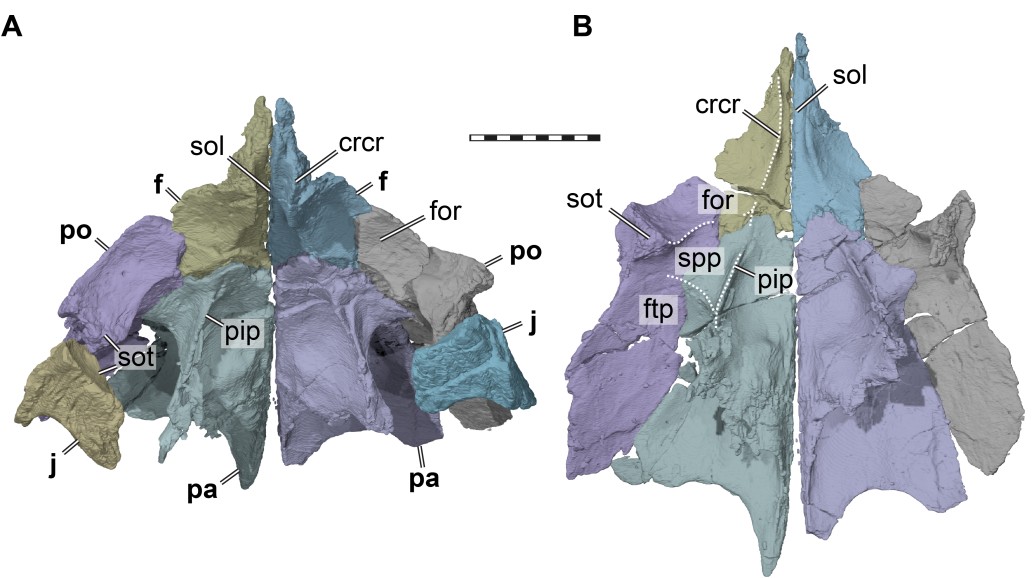

**Figure 3** **Three dimensional renderings of the orbitotemporal region of *Pleurosternon bullockii* (UMZC T1041).** (A) Anteroventral view of partial cranium. (B) Ventral view of partial cranium. Abbreviations: crcr, crista cranii; f, frontal; ftp, fossa temporalis; j, jugal; for, fossa orbitalis; pa, parietal; pip, processus inferior parietalis (descending process of parietal); po, postorbital; sol, sulcus olfactorius; sot, septum orbitotemporale; spp, sulcus palatino-pterygoideus. Scale bar equals 10 mm.

covered the supraoccipital entirely (see supraoccipital). The posterior margin of the parietal is emarginated lateral to the midline, but still conceals most of the otic capsule in dorsal view (Fig. 2A), as is also the case in *Dorsetochelys typocardium* (*Evans & Kemp, 1976*) and *Uluops uluops* (UCM 53971). The right parietal of UMZC T1041 is broken posterolaterally near the squamosal, so that the contact between these bones is not preserved. However, if the right squamosal were a mirror image of the one on the left side, where the squamosal is missing but the parietal completely preserved, both bones would clearly have a short contact posterior to the postorbital. A squamosal-parietal contact is also present in *Dorsetochelys typocardium* (*Evans & Kemp, 1976*) and *Uluops uluops* (UCM 53971).

The descending process of the parietal is well-developed in UMZC T1041 (Figs. 3 and 4B), although the anterior margin of the process is posteriorly retracted, so that the interorbital fenestra is large. Internally, the medial surface of the base of the descending process seems bulged within the endocranial cavity, suggesting that *Pleurosternon bullockii* possibly had large cerebral hemispheres. The posteroventral margin of the descending process of the parietal contacts the supraoccipital posteriorly and the prootic anteriorly. At the dorsal margin of the trigeminal foramen, the parietal forms a thin posteroventral process that also frames the posterior margin of the foramen, and extends to contact the pterygoid, thereby excluding the prootic from contributing to the foramen (Fig. 4). This is visible on the right side of UMZC T1041, where the prootic, parietal, quadrate, and partially the pterygoid and epipterygoid are preserved in articulation. The same arrangement can be observed in *Uluops uluops* (UCM 53971) at the posterior margin of the trigeminal foramen. A posteroventral process of the parietal along the posterior trigeminal foramen

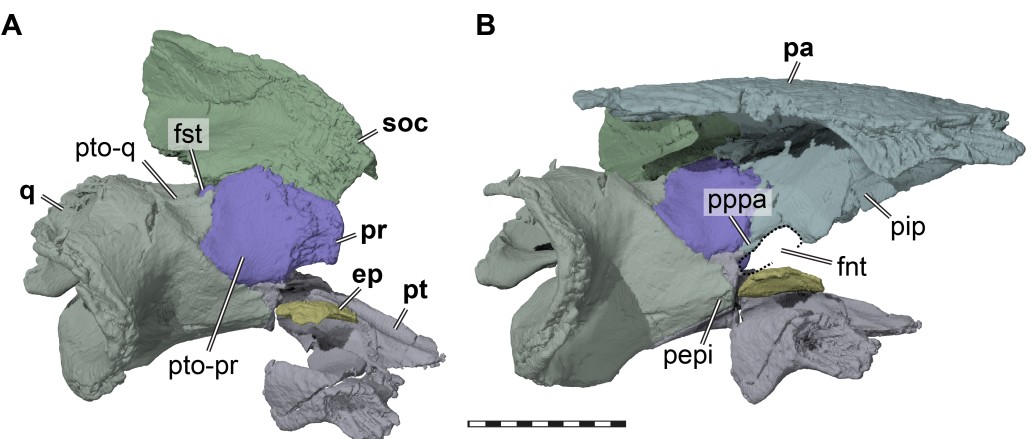

**Figure 4** Three dimensional renderings of the trigeminal region of *Pleurosternon bullockii* (UMZC T1041). (A) Anterodorsolateral view of partial right cranium. (B) Anterolateral view of partial right cranium. Abbreviations: ep, epipterygoid; fnt, foramen nervi trigemini (trigeminal foramen); fst, foramen stapedio-temporale; pa, parietal; pepi, processus epipterygoideus; pip, processus inferior parietalis (descending process of parietal); pppa, posterior process of parietal behind trigeminal foramen; pr, prootic; pt, pterygoid; pto-pr, prootic part of processus trochlearis oticum; pto-q, quadrate part of processus trochlearis oticum; q, quadrate; soc, supraoccipital. Scale bar equals 10 mm.

margin has been proposed as a synapomorphy of Thalassochelydia (*Anquetin, Püntener & Joyce, 2017*), based on the presence of this process in *Solnhofia parsonsi* and plesiochelyids (*Anquetin, Püntener & Billon-Bruyat, 2015*). The same process has also been observed in sandownids (*Evers & Joyce, 2020*). The presence of this process in paracryptodires as well as the aforementioned groups, all of which have recently been interpreted as stem-turtles (e.g., *Evers & Benson, 2019*), indicates that this could be a feature that is more widespread along the crownward stem-lineage of turtles. The contacts of the parietal of UMZC T1041 anteroventral to the trigeminal foramen are not clearly preserved on either side of the specimen. However, contacts with both the epipterygoid and pterygoid were likely present.

Most of the lateral contact with the postorbital is formed as a simple, planar contact without further reinforcement. However, at the level of the base of the descending process, the parietal expands laterally underneath the postorbital (Fig. 3). The ventral surface of this buttress is hypertrophied to a mediolateral ridge between the descending process of the parietal and the ventral process of the postorbital. This shallow ridge forms the posterior boundary of a shallow fossa that is anteriorly bound by the medial margin of the orbital fossa, and which we identify as as the roof to the sulcus palatino-pterygoideus (see postorbital). The same fossa is also present in various other turtle, though rarely reported, we note that it is particularly deep in UMZC T1041, *Uluops uluops* (UCM 53971), and pleurodires (*Gaffney, Tong & Meylan, 2006*).

**Postorbital.** Both postorbitals are preserved in UMZC T1041 (Fig. 2). The postorbital contacts the frontal anteromedially, the parietal posteromedially, the squamosal posteriorly, the quadratojugal posteroventrally, and the jugal ventrally.

The postorbital of UMZC T1041 is posteriorly elongate (Figs. 2A–2D) and forms the extensive temporal roofing together with the parietal. However, the postorbital is excluded from the posterior margin of the skull roof by a squamosal-parietal contact. The postorbital and squamosal contact is located dorsal to the margin of the cavum tympani, but separated from contacting the cavum or quadrate by a posterior process of the quadratojugal (Fig. 2D). The quadratojugal contacts the postorbital posterior to the jugal-postorbital contact.

The postorbital forms the posterior margin of the orbit (Figs. 2C–2D). Although the preservation of UMZC T1041 leaves a bit of room for interpretation regarding the posterior orbital margin, the jugal was excluded clearly from contributing to the margin of the orbit by a short ventral process of the postorbital that contacted the maxilla (see jugal). Within the orbital fossa, the postorbital forms an expanded ventral process that contacts the medial process of the jugal ventrally (Fig. 3). Hereby, the ventral process of the postorbital nearly contacts the pterygoid, and a contact with the unpreserved palatine cannot be ruled out entirely. The morphology of the ventral process of the postorbital seems to be very similar in *Uluops uluops* (UCM 53971). In this taxon, the postorbital comes close to the pterygoid and palatine, but both contacts are absent. A postorbital-palatine contact is only seen in pleurodires (*Gaffney, 1979b*). However, the orbito-temporal region of both UMZC T1041 and *Uluops uluops* (UCM 53971) show striking similarities to the morphology of that region in pleurodires: the medial postorbital-jugal expansion forms an extended posterior wall to the orbital fossa (Fig. 3). Additionally, again as in pleurodires, the passage between the temporal fossa posteriorly, and the orbital fossa anteriorly, becomes strongly constricted between the ventral process of the postorbital and the descending process of the parietal (Fig. 3). Furthermore, the roof of this constricted trough is depressed to form a shallow fossa (Fig. 3). This morphology is otherwise only found in pleurodires (e.g., *Gaffney, Tong & Meylan, 2006*). Thus, we use terms from the pleurodire literature to refer to these respective structures; we use 'septum orbitotemporale' for the medial expanded jugal-postorbital wall (*Gaffney, Tong & Meylan, 2006*: fig25), and we use 'sulcus palatino-pterygoideus' for the space between the orbital and temporal fossae (*Gaffney, Tong & Meylan, 2006*): fig25, although we note that it is incipient with respect to the pleurodire morphology.

**Jugal.** Both jugals of UMZC T1041 are preserved (Figs. 2A–2D), but the elements are disarticulated slightly from their surrounding bones. Definite contacts are with the postorbital, maxilla, quadratojugal, and pterygoid, and a contact with the palatine cannot be ruled out. The jugal of UMZC T1041 shows typical turtle morphology with a laterally placed, vertical plate that forms part of the lateral skull roof, and a medially directed process that contacts parts of the palate. The jugal is removed from the ventral skull margin, as the maxilla forms a posterior process that extends ventrally underneath the jugal for some distance (Figs. 2C–2D). The end of this maxillary process forms the beginning of the moderately deep cheek emargination, which extends from the maxilla along the ventral margins of the jugal and quadratojugal. The dorsally positioned jugal and step-like margin between the maxilla and jugal are also present in *Uluops uluops* (UCM 53971), and baenids

(*Gaffney, 1972*), but not so in *Dorsetochelys typocardium* (*Evans & Kemp, 1976*), in which the jugal is the posterior continuation of the maxilla along the labial margin of the skull.

It seems that the jugal was excluded from the orbital margin, as is also the case in *Glyptops ornatus* (*Gaffney, 1979a*), *Arundelemys dardeni* (*Lipka et al., 2006*), and an assortment of advanced baenids (*Gaffney, 1972*; *Joyce & Lyson, 2015*; *Rollot, Lyson & Joyce, 2018*) as well as helochelydrids (e.g., *Joyce et al., 2011*; *Joyce, Sterli & Chapman, 2014*): on both sides, the jugal of UMZC T1041 has an anteriorly recessed surface, which is posteriorly delimited by a rim. Only posteriorly to this rim does the surface of the bones show the sculpturing that covers most of the external bone surfaces (Figs. 2C–2D). In *Uluops uluops* (UCM 53971), in which the jugal contributes to the orbit, the surface sculpturing of the jugal bone indeed extends to the orbit. Thus, we interpret the recessed surface in UMZC T1041 to be a laterally exposed facet for the overlying postorbital and maxilla. Our interpretation hereby differs markedly from that of *Evans & Kemp (1975)*, who reconstructed a point contact of the jugal to the orbit margin. Despite its likely exclusion from the orbital margin itself, the jugal of UMZC T1041 is expressed medially to the orbital margin within the orbital fossa, which we call septum orbitotemporale (see postorbital) (Fig. 3). Extending posteromedially from the orbital fossa, the medial process of the jugal reaches the pterygoid. This section of the jugal is ventrally overlain by the maxilla. A contact with the palatine might have been present in this area, but this cannot be confirmed without any indication of the morphology of the palatine bone.

**Quadratojugal.** The quadratojugal is only preserved on the right side of UMZC T1041 (Fig. 2D). The quadratojugal is triradiate, as it forms a posterior process that inserts between the postorbital and quadrate to contact the squamosal, extends ventrally along the anterior quadrate margin, and anteriorly towards the jugal. The quadratojugal of *Pleurosternon bullockii* is a comparatively small bone in comparison to those of basal stem-turtles (e.g., *Gaffney, 1990*) or baenids (e.g., *Joyce & Lyson, 2015*; *Rollot, Lyson & Joyce, 2018*), but not as small as in sichuanchelyids (*Sukhanov, 2000*; *Joyce et al., 2016*), which have quadratojugals with a greatly reduced lateral exposure. The quadratojugal of UMZC T1041 is similar in size and extent to that of *Uluops uluops* (UCM 53971). In UMZC T1041, the ventral margin of the quadratojugal contributes to the moderately deep cheek emargination. Although the quadratojugal frames the quadrate anteriorly, it is slightly displaced anteriorly from the margin of the cavum tympani, and clearly does not closely approach the articular condyles of the quadrate.

**Squamosal.** Only the right squamosal of UMZC T1041 is preserved (Fig. 2). Contacts with the quadratojugal, quadrate, and opisthotic can be documented, and a parietal contact is inferred. The squamosal of *Pleurosternon bullockii* is a large element with a well-developed posterior process and a deeply recessed antrum postoticum. The squamosal caps the posterior part of the quadrate, and hereby forms the posterodorsal margin of the deeply developed cavum tympani, as is also the case in *Uluops uluops* (UCM 53971), *Dorsetochelys typocardium* (*Evans & Kemp, 1976*), *Compsemys victa* (*Lyson & Joyce, 2011*), and baenids (*Joyce & Lyson, 2015*). The squamosal of UMZC T1041 has a deep, incision-like facet for the opisthotic, and thus frames the paroccipital process from dorsal and ventral. Whereas the

anterior part of the squamosal is bulbous and houses the expanded antrum postoticum, the bone narrows mediolaterally toward its posterior end to form a vertically sheeted process that extends posteriorly just beyond the level of the supraoccipital crest (Fig. 2A, Fig. 2F).

**Premaxilla.** Both premaxillae are preserved in UMZC T1041 (Figs. 2A–2E). The premaxilla contacts the maxilla, but contacts with the palatine are uncertain due to incomplete preservation. A contact with the vomer was originally preserved (*Evans & Kemp, 1975*), but the vomer is now lost (see vomer). The right premaxilla completely surrounds a clearly developed foramen praepalatinum (Fig. 2B), which is only incompletely preserved on the other side. The dorsal margin of the premaxilla shows that the external naris was undivided from below in *Pleurosternon bullockii*, as described by *Evans & Kemp (1975)*. Within the nasal fossa, both premaxillae together form a low ridge that probably serves as the insertion for the internarial septum. The full labial ridge, including the premaxillae, frame a broad notch along the midline. Serrations on the premaxillary labial ridge, as reported by *Evans & Kemp (1975)*, are not confirmed for UMZC T1041.

**Maxilla.** Both maxillae are preserved in UMZC T1041 (Figs. 2A–2E). The maxilla contacts the premaxilla anteriorly, the nasal and prefrontal dorsally, the jugal and postorbital posteriorly, and the pterygoid posteromedially. A medial contact with the vomer was previously documented by *Evans & Kemp (1975)*, but is not preserved anymore (see section on the vomer). A contact with the palatine was likely present along the medial margin of the triturating surface. The triturating surface of the maxilla is relatively narrow and lacks a medial ridge (Fig. 2B). The labial ridge is ventrally deep and sharp-edged (Fig. 2B). In anterior view, the labial ridge of the maxilla and premaxilla jointly frame a broad, but shallow median notch (Fig. 2E). A curved labial margin of the maxilla seems common among paracryptodires, although the degree of the curvature varies. UMZC T1041 is intermediate in this regard between *Glyptops ornatus* (*Gaffney, 1979a*), *Dorsetochelys typocardium* (*Evans & Kemp, 1976*), and *Arundelemys dardeni* (*Lipka et al., 2006*) with a weakly curved maxilla on one side, and *Uluops uluops* (UCM 53971) on the other, in which the curvature is very strongly pronounced. The medial margin of the incompletely preserved foramen orbito-nasale of UMZC T1041 is formed by the maxilla, and the foramen alveolaris superior is a comparatively large foramen positioned within that margin (best viewed in the 3D models). Posterior to the ascending process, the maxilla forms the complete ventral margin of the orbit (Figs. 2C–2D). The margin is slightly raised to an orbital rim, medial to which the maxilla forms a narrow floor of the orbital fossa. The foramen supramaxillare is small and positioned immediately anterior to the jugal contact within the floor of the orbital fossa (*Evans & Kemp, 1975*). Posteriorly, the maxilla laterally overlaps the jugal to contact the postorbital along the posterior margin of the orbit (see jugal). The maxilla also extends ventrally to the jugal with a short process (Figs. 2C–2D). The posterior part of the maxilla is medially expanded underneath the jugal and reaches the anterior end of the external process of the pterygoid.

**Vomer.** *Evans & Kemp (1975)* describe and illustrate a small anterior part of the vomer that was articulated with the premaxillae and the prefrontals. However, this part of the
vomer, alongside a small posterior part of the left premaxilla is not preserved in the specimen anymore, and the whereabout of the fragments that must have broken off since the description of *Evans & Kemp (1975)* are unclear.

**Palatine.** The palatines are not preserved in UMZC T1041.

**Quadrate.** Both quadrates of UMZC T1041 are preserved (Figs. 2B–2D, 2F). The quadrate contacts the prootic and opisthotic medially, the pterygoid anteroventrally, the epipterygoid anteriorly, the quadratojugal anterolaterally, and the squamosal posterolaterally. A point contact with the supraoccipital is likely just posterior to the foramen stapedio-temporale. The lateral surface of the quadrate is dominated by the deeply recessed cavity that forms most of the cavum tympani (Figs. 2C–2D). The quadrate forms the anterior margin of the cavum tympani, but posterolaterally this structure is formed by the squamosal. The incisura columella auris is posteroventrally open, as in all other paracryptodires (*Joyce & Lyson, 2015*; *Joyce & Anquetin, 2019*). The articular process of the quadrate of UMZC T1041 does not protrude ventrally very deep and does not significantly expand beyond the ventral margin of the cavum tympani. Much of the lateral surface of the articular process was probably covered by the ventral ramus of the quadratojugal. In ventral view, the articular surface of the process bears two weakly convex sub-facets (Fig. 2B), which are mediolaterally separated by a shallow sulcus (*Evans & Kemp, 1975*). The medial side of the articular process is buttressed by the posterior process of the pterygoid, but the process does not reach the articular facet (Fig. 2B). Medially, within the cavum acustico-jugulare, the quadrate forms together with the pterygoid and prootic the posterior foramen for the canalis cavernosus. Between the prootic and quadrate, the canalis stapedio-temporalis connects the cavum acustico-jugulare with the adductor chamber (Fig. 5A). The respective exiting foramen, the foramen stapedio-temporale, is formed on the dorsal surface of the otic capsule at the posterior limit of the contact between the quadrate and prootic (Fig. 4A). The foramen and associated canal have a relatively small diameter, superficially similar to trionychians or kinosternoids. However, the diameters of the stapedial and cerebral canals are about equally sized in UMZC T1041, whereas the cerebral artery canal is much larger than the stapedial artery canal in the aforementioned cryptodires. Immediately posterior to the foramen stapedio-temporale of UMZC T1041, the quadrate likely has a short, superficial contact with the supraoccipital, which can also be observed in *Uluops uluops* (UCM 53971) and some baenids like *Stygiochelys estesi* (*Gaffney, 1972*). Contacts of the quadrate with the opisthotic and squamosal form the posterior aspect of the floor of the adductor chamber in UMZC T1041. Anteriorly within the adductor chamber, the dorsal surface of the quadrate contributes to the formation of the weakly developed processus trochlearis oticum. However, the quadrate forms less of the process than the prootic. The process is developed as a shallowly concave area along the quadrate-prootic contact (Fig. 4A). The processus trochlearis oticum of *Uluops uluops* (UCM 53971) is somewhat more pronounced, but otherwise similar to that of *Pleurosternon bullockii*. The anterior quadrate surface of UMZC T1041, ventral to the processus trochlearis oticum, extends

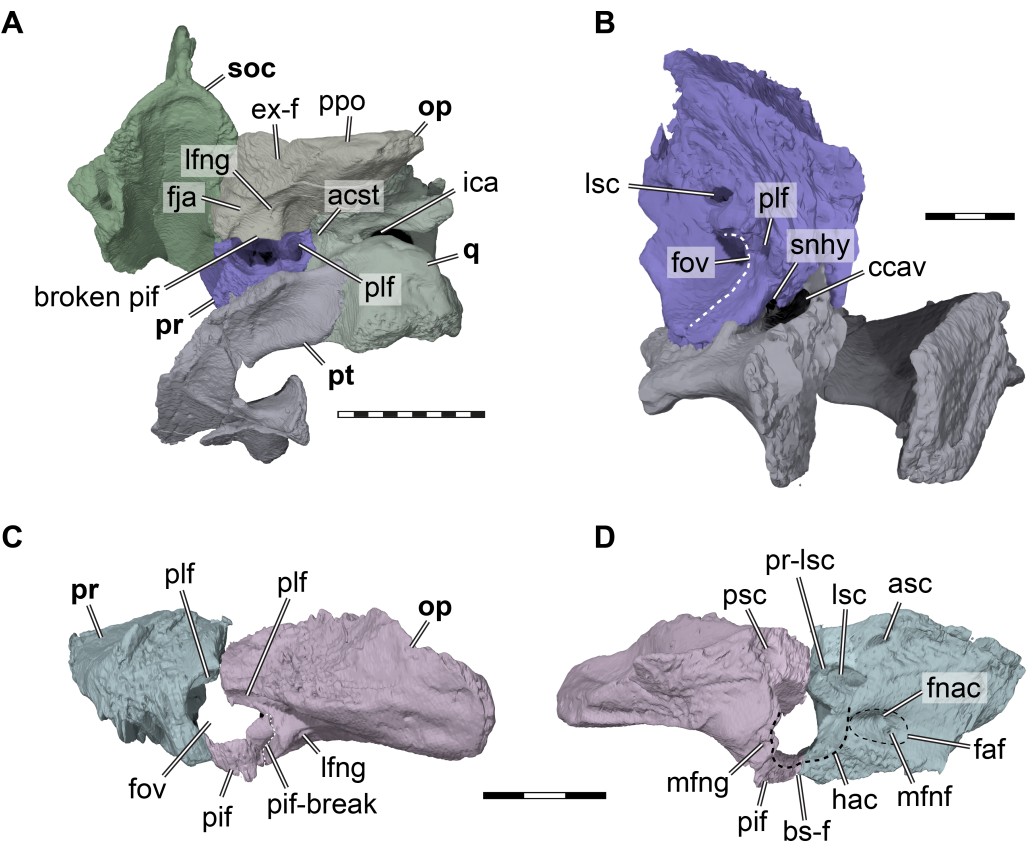

**Figure 5** Three dimensional renderings of aspects of the cavum acustico-jugulare and the inner ear capsule of *Pleurosternon bullockii* (UMZC T1041). (A) Posteroventral view of partial right basicranium. (B) Posterolateral view of right prootic and pterygoid. (C) Lateral view of left prootic and opisthotic. (D) Medial view of left prootic and opisthotic. Abbreviations: acst, aditus canalis stapedio-temporalis; asc, anterior semicircular canal; bs-f, parabasisphenoid-facet; ccav, canalis cavernosus; ex-f, exoccipital facet; faf, fossa acustico-facialis; fja, foramen jugulare anterius; fnac, foramen nervi acustici; fov, fenestra ovalis; hac, hiatus acusticus; ica, incisura columella auris; lfng, lateral foramen nervi glossopharyngei; lsc, lateral semicircular canal; mfng, medial foramen nervi glossopharyngei; mfnf, medial foramen nervi facialis; op, opisthotic; pif, processus interfenestralis; plf, perilymphatic fossa; ppo, paroccipital process; pr, prootic; pr-lsc, prootic-part of the lateral semicircular canal; psc, posterior semicircular canal; pt, pterygoid; q, quadrate; snhy, sulcus nervi hyomandibularis; soc, supraoccipital. Scale bars equals 10 mm in A, 3 mm in B–D.

anteriorly in a short epipterygoid process (Fig. 4A). This process contacts the epipterygoid but is excluded from the trigeminal foramen margin by a dorsal process of the pterygoid.

**Epipterygoid.** In UMZC T1041, we identify a small plate-like ossification ventral to the trigeminal foramen as the epipterygoid and thus agree with *Evans & Kemp (1975)* in the identification of this element. However, the epipterygoids are difficult to segment in our CT scans, due to their highly interdigitated sutures with the parietals, and fragmentation due to breakage in the respective skull area. On the right side of the specimen, the epipterygoid seems to be preserved close to its original position, although its dorsal margin has tipped laterally (Fig. 4). Here, the epipterygoid is positioned anteriorly to the epipterygoid process

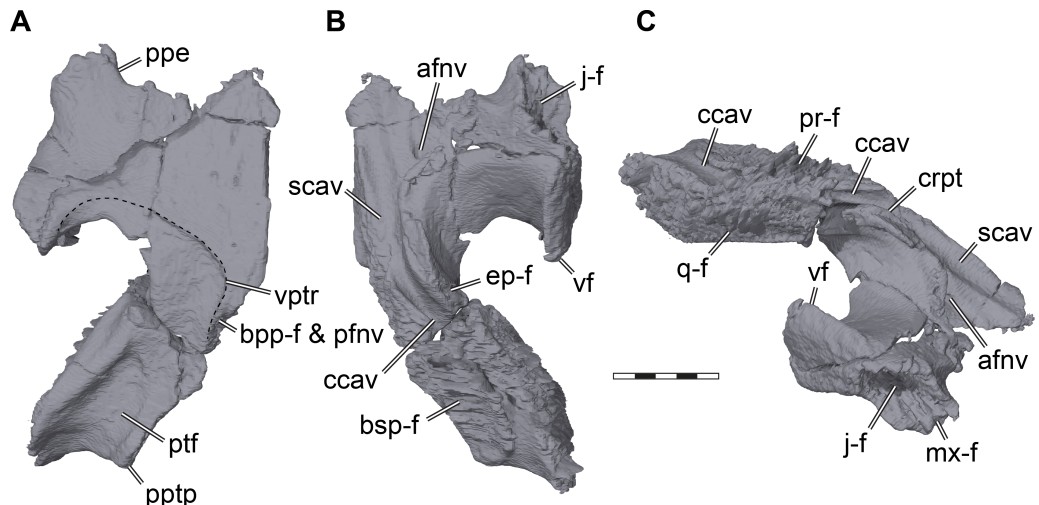

**Figure 6 Three dimensional renderings of the right pterygoid of *Pleurosternon bullockii* (UMZC T1041).** (A) Ventral view. (B) Dorsal view. (C) Anterodorsolateral view. Abbreviations: afnv, anterior foramen for the canalis nervus vidiani; bpp-f, basisptergyoid process facet; bsp-f, parabasisphenoid facet; ccav; canalis cavernosus; crpt, crista pterygoidei; ep-f, epipterygoid facet; j-f, jugal facet; mx-f, maxilla facet; pfnv, posterior foramen for the canalis nervus vidiani; ppe, processus pterygoideus externus; pptp, posterior pterygoid process; pr-f, prootic facet; ptf, pterygoid fossa; q-f, quadrate facet; scav, sulcus cavernosus; vf, vertical flange of the external pterygoid process; vptr, ventral pterygoid ridge. Scale bar equals 3 mm.

of the quadrate and ventrally to the descending process of the parietal. If the epipterygoid were tilted back into contact with the parietal, it would frame the anteroventral margin of the trigeminal foramen, its usual position (e.g., *Gaffney, 1979b*). The left epipterygoid, which seems to have been misplaced slightly dorsally, also indicates that a contact with the prootic is likely.

**Pterygoid.** Both pterygoids of UMZC T1041 are preserved (Fig. 2B). The right pterygoid is basically complete (Fig. 6), but has a transverse break through its center, whereas in the left pterygoid, the external process is broken off but preserved. The pterygoid contacts the parabasisphenoid, jugal, maxilla, parietal, epipterygoid, quadrate, prootic, opisthotic. Although the palatines and the basioccipital are absent, contacts with the pterygoid seem all but certain. However, as the exoccipitals are missing as well, we cannot assess if a pterygoid contact was present with this bone as well. An interpterygoid contact below the rostrum basisphenoidale is certainly absent (see Parabasisphenoid), but a possible midline contact anterior to the parabasisphenoid cannot be ruled out, as the anterior tips of the pterygoids might be damaged.

The pterygoid of *Pleurosternon bullockii* shows a mixture of plesiomorphic and derived features, as well as similarities to pleurodires and cryptodires. The general architecture of the bone mimics that of most cryptodires, with a relatively long posterior process that covers large parts of the cavum acustico-jugulare from ventral exposure (Figs. 2B, 5A). The posterior process extends posteriorly up to the tips of the posterior parabasisphenoid processes (= anterior tubercula basioccipitale; see parabasisphenoid), and certainly

contacted the basioccipital, as is also the case in *Uluops uluops* (UCM 53971). The posterior pterygoid process of UMZC T1041 furthermore completely covers the ventral end of the processus interfenestralis of the opisthotic (Figs. 2B, 5A), as in *Uluops uluops* (UCM 53971) and *Glyptops ornatus* (*Gaffney, 1979a*). Thus, the process is much more cryptodire-like and extensive than acknowledged in previous descriptions (*Evans & Kemp, 1975*), although *Gaffney (1979a)* corrected earlier statements of his, and also argued that in both *Glyptops ornatus* and UMZC T1041 the posterior process of the pterygoid is indistinguishable from cryptodires. Between the articular process of the quadrate and the parabasisphenoid of UMZC T1041, the posterior process of the pterygoid shows a well-developed pterygoid fossa (Figs. 6A, 7C). Despite this cryptodire-like appearance, the pterygoid retains a deep, pocket-like facet on its medial margin for the reception of the basipterygoid processes of the parabasisphenoid (Figs. 6A, 7C). This facet is absent in the crania of all Recent turtles. The medial margin of the pterygoid is laterally curved around the basipterygoid articulation, which results in the formation of a window-like recess in the floor of the basicranium, through which the basipterygoid process of the parabasisphenoid is partially exposed, and through which the cerebral artery and the vidian nerve enter the cranium. The articulation socket for the basipterygoid process within the medial surface of the pterygoid bears a central foramen, which is positioned between the dorsal and ventral flanges of the basipterygoid process (see Parabasisphenoid). The foramen is the entry to the canalis nervus vidianus, which transmits the vidian nerve through the pterygoid towards an anterior foramen that is positioned lateral to the anterior end of the crista pterygoidei (Fig. 6B). The connection of these two foramina and their identification as the canalis nervus vidianus were already correctly inferred by *Evans & Kemp (1975)*, but the foramen within the basipterygoid facet was re-interpreted by *Sterli et al. (2010)* as the posterior entry foramen of the palatine artery. Our CT scans unambiguously confirm the internal connection of the above-mentioned foramen with the anterior foramen of the canalis nervus vidianus, and thus corroborate the interpretation by *Evans & Kemp (1975)*. *Gaffney (1979a)* described the presence of a palatine artery canal for *Glyptops ornatus* that extends anteriorly from within the basipterygoid articulation socket of the pterygoid, but he could not find an anterior exiting foramen. Thus, it is possible that *Gaffney (1979a)* misinterpreted the foramen he saw, which could instead be for the vidian nerve canal, and did not find an anterior palatine artery canal foramen, because it does not exist. However, this can only be tested with certainty by attaining CT scans of *Glyptops ornatus*. For UMZC T1041, we can furthermore confirm the canalis pro ramo nervi vidiani of *Evans & Kemp (1975)*; foramen pro ramo nervi vidiani prior to revision of *Rollot, Lyson & Joyce (2018)*), as a vertically oriented canal that extends through the pterygoid from the floor of the canalis cavernosus to the ventral pterygoid surface posterior to the parabasisphenoid articulation (Fig. 7C). The canalis pro ramo nervi vidiani seems to be present in the same location in *Glyptops ornatus* (*Gaffney, 1979a*). Immediately anterior to the ventral opening of the canalis pro ramo nervi vidiani of UMZC T1041, a robust ridge surrounds the ventromedial margin of the pterygoid around the basipterygoid facet (*Evans & Kemp, 1975*). Both the ventral canalis pro ramo nervi vidiani and the robust ridge are affected by the large break of the right pterygoid, but nicely preserved on the left element (Fig. 7C). The ridge persists

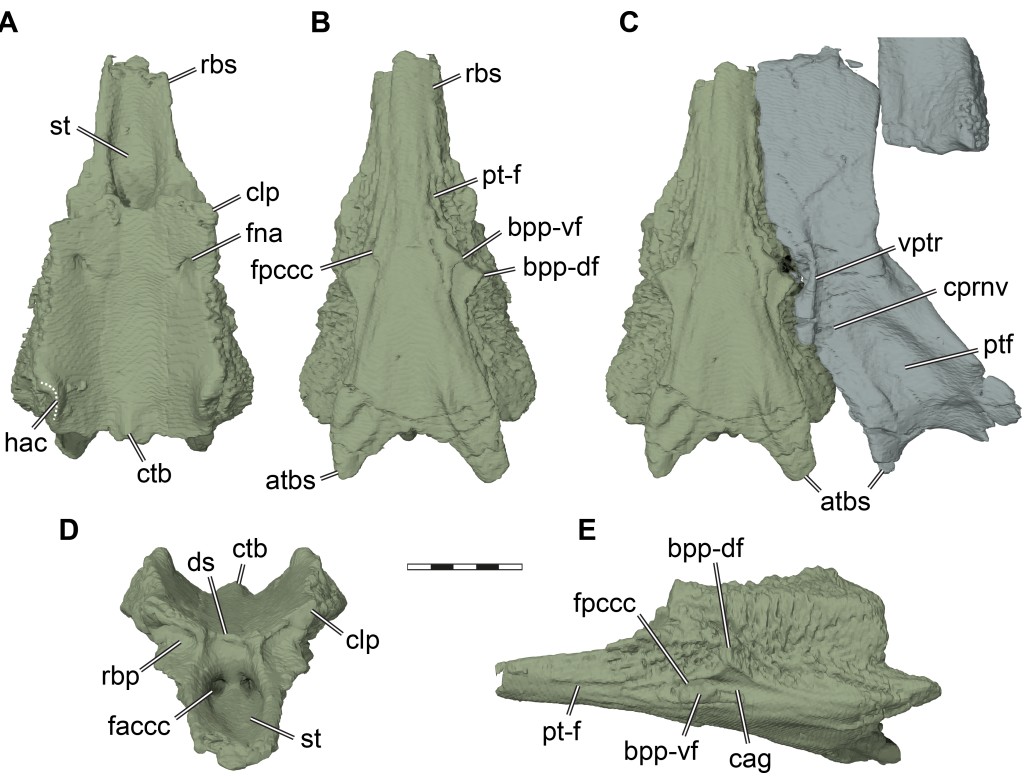

**Figure 7** **Three dimensional renderings of the parabasisphenoid and basipterygoid process of Pleurosternon bullockii (UMZC T1041).** (A) Parabasisphenoid in dorsal view. (B) Parabasisphenoid in ventral view. (C) Parabasisphenoid and left pterygoid in ventral view. (D) Parabasisphenoid in anterodorsal view. (E) Parabasisphenoid in left lateral and slight ventral view. Abbreviations: atbs, anterior tubercula basisphenoidale; bpp-df, dorsal flange of basipterygoid process; bpp-vf, ventral flange of basipterygoid process; cag, carotid groove; clp, clinoid process; ctb, crista tuberculi basalis; ds, dorsum sellae; faccc, foramen anterius canalis caroticus cerebralis; fna, foramen nervi abducentis; fpccc, foramen posterius canalis caroticus cerebralis; fprnv, foramen pro ramo nervi vidiani; pt-f, pterygoid facet; ptf, pterygoid fossa; rbp, retractor bulbi pit; rbs, rostrum basisphenoidale; st, sella turcica; vptr, ventral pterygoid ridge. Scale bar equals 3 mm.

anteriorly, albeit much more shallowly, parallels the parabasisphenoid suture for a short distance, and curves laterally toward the posteriorly directed tip of the external pterygoid process (Fig. 6A). *Evans & Kemp (1975)* interpret the ridge to be the insertion for the pterygoideus musculature, and the same ridge morphology is present in *Glyptops ornatus* (*Gaffney, 1979a*). In *Uluops uluops* (UCM 53971), the pterygoideus muscle ridge is similar, but the ridge traverses the pterygoid earlier and extends to the posterior end of the external process, thus defining a smaller fossa.

The external process of the pterygoid is extremely large in UMZC T1041 and projects laterally deeply into the subtemporal fenestra (Figs. 2B, 5A–5B, 6). Although many turtles, including many cryptodires (e.g., chelydrids, emydids, geoemydids), have their lateral surface of the external process developed to a vertical flange, this morphology seems hypertrophied in *Pleurosternon bullockii*. The vertical plate of the lateral surface of the external process is less sheeted than in most cryptodires that have the respective flange,

but is more robust, and notably gently medially deflected towards its dorsal margin (Figs. 6B–6C, 5A–5B, 4). As such, the morphology of the external process of the pterygoid of UMZC T1041 comes very close to that of a pleurodiran processus trochlearis pterygoidei. Additionally, although the external process of the pterygoid is anteriorly sutured to the jugular-maxillar area in many turtles, including some cryptodires (chelydrids) and early stem turtles like *Proganochelys quenstedtii* (e.g., *Gaffney, 1990*), the deep and large facet for the articulation with the jugal (Figs. 6B–6C, 4) demonstrates an unusually strong structural integration of the external process into the mechanical framework of the cranium, as is also seen in pleurodires. However, the lateral surface of the external process is parallel to the sagittal skull plane (Fig. 2B), and the process is not laterally tilted with its posterior end, as is the case in pleurodiran pterygoid trochleae. As such, the trajectory of the adductor muscles, when approximated around the weak otic trochlea present on the prootic of UMZC T1041, do not seem to be guided around the external process, implying functional differences in the structure described for *Pleursternon bullockii* and the processus trochlearis pterygoidei of pleurodires. As the detailed morphology of the external process is rarely described in detail, we cannot exclude the possibility that this morphology is perhaps wider spread that noted herein.

The dorsal surface of the pterygoid of UMZC T1041 is curved around the anteriorly narrowing parabasisphenoid. Posteriorly, the medial part of the dorsal surface is dominated by a low buttress for articulation with the prootic and parabasisphenoid, which also forms the medial wall of the canalis cavernosus (Figs. 6B–6C). The lateral surface of this canal is formed by the crista pterygoidei. The canalis cavernosus ends anteriorly in the foramen cavernosum formed between the prootic and pterygoid. Anterior to the foramen, the canalis cavernosus continues as the sulcus cavernosus and parallels the rostrum basisphenoidale of the parabasisphenoid (Figs. 6B–6C). At the anterior end of the crista pterygoidei, there is a small exiting foramen for the canalis nervus vidianus (*Evans & Kemp, 1975*; Figs. 6B–6C). The crista pterygoidei itself is relatively low and posteriorly interrupted by a deep notch for the trigeminal foramen. In the posterior margin of this foramen, the pterygoid extends dorsally to reach the parietal and thus excludes the prootic from contributing to the foramen (Fig. 4B). The same dorsal process is present in *Uluops uluops* (UCM 53971). The crista pterygoidei of UMZC T1041 is laterally overlapped by the epipterygoid anterior to the trigeminal foramen (Fig. 4B).

**Supraoccipital.** The supraoccipital of UMZC T1041 is nearly completely preserved. It ventrolaterally contacts the prootic and opisthotic and perhaps has a point contact with the quadrate (Figs. 2 and 4). It furthermore contacts the parietal anteroventrally and dorsally, and likely contacted the exoccipital posteroventrally.

The supraoccipital of UMZC T1041 forms a comparatively broadly arched roof to the endocranial cavity (Fig. 2F). A vertical plate extends over the entire anteroposterior length of the supraoccipital, but this plate does not seem to extend posteriorly from the approximate position of the foramen magnum as a supraoccipital crest (*Evans & Kemp, 1975*). This crest seems dorsally nearly completely covered by the parietals. The supraoccipital was likely only marginally visible in the skull roof between the parietals, though much less than interpreted

by *Evans & Kemp (1975)*. This differs from the conditions in *Uluops uluops* (UCM 53971), *Glyptops ornatus* (*Gaffney, 1979a*), and *Dorsetochelys typocardium* (*Evans & Kemp, 1976*), in which the supraoccipital crest is topped by a triangular or diamond shaped horizontal plate that forms a marginal part of the dorsal skull roof. Within the otic capsule of UMZC T1041, the supraoccipital articulates with the prootic anteriorly and the opisthotic posteriorly. Between both bones, the supraoccipital likely extends laterally to have a small superficial contact with the quadrate, as is also the case in *Uluops uluops* (UCM 53971). The hiatus acusticus of UMZC T1041 is a large, unossified space between the supraoccipital, prootic, parabasisphenoid, and opisthotic, leaving a large opening between the inner ear cavity and the endocranial cavity. Dorsally above the hiatus acusticus, the foramen aquaducti vestibuli, i.e., the canal for the endolymphatic duct, is completely embedded within bone, forming a short canal from the endocranial cavity into the recess of the supraoccipital that holds the common crus of the labyrinth. *Evans & Kemp (1975)* reported that the foramen aquaducti vestibuli is only a notch in the supraoccipital margin, but as completely ossified canals exist on both sides of the supraoccipital, they probably misinterpreted a small break on the left side of the element and missed the small foramina. Contacts with the lost exoccipitals can be inferred from short facets posterior to the articulation site for the opisthotic.

**Exoccipital.** The exoccipitals are not preserved in UMZC T1041.

**Basioccipital.** The basioccipital is not preserved in UMZC T1041.

**Prootic.** Both prootics of UMZC T1041 are present, but the right one is better preserved. The prootic has a large exposure within the floor of the adductor chamber, forming the anteromedial part of the otic capsule (Fig. 4). Its dorsal surface is transversely concave and anteroventrally sloping toward the subtemporal fossa (Fig. 4A). Although this surface does not protrude anteriorly into the fossa, this morphology clearly indicates a redirecting of the jaw adductor musculature. Thus, a small processus trochlearis oticum is present. A very similar morphology, although more pronounced by a deeper concave flexure across the prootic, is seen in *Uluops uluops* (UCM 53971). Anteroventrally, the prootic of UMZC T1041 forms the dorsal margin of the foramen cavernosus. Slightly anterodorsally to the position of this foramen, the prootic is slightly expanded transversely to form the posterior wall of the fossa epiptericum for the trigeminal ganglion. However, the anteroventral margin of the prootic is excluded from the trigeminal foramen itself, contra the description of *Evans & Kemp (1975)*; Fig. 4B). Medioventrally, the prootic has a large, foot-like process that contacts the pterygoid and parabasisphenoid. The medial surface of this ventral prootic process is recessed by the fossa acustico-facialis, from which the facial nerve canal extends through the ventral prootic process toward the canalis cavernosus, and from which the short acustic nerve canals lead into the prootic part of the cavum labyrinthicum. This cavity for the labyrinth deeply excavates the prootic from posterior (Figs. 5A–5B). The canal for the lateral semicircular duct of the labyrinth organ is already ossified within the prootic, and thus shared between prootic and opisthotic (Figs. 5B, 5D). Ventrally, a horizontal footplate of the ventral prootic process extends posteriorly and floors the anterior half of the cavum labyrinthicum (Fig. 5). This footplate contacts the opisthotic

across the entire transverse width of the cavum labyrinthicum, so that the fenestra ovalis is completely surrounded, dorsally and ventrally, by these two bones (Figs. 5C–5D). The same condition is present in *Uluops uluops* (UCM 53971) and *Glyptops ornatus* (*Gaffney, 1979a*). Laterally to the fenestra ovalis of UMZC T1041, and thus within the cavum acustico-jugulare, the posteriorly exposed surface of the prootic is deeply recessed by a fossa (Figs. 5B–5C), which has been hypothesized to be related to the perilymphatic system of reentrant fluid-flow systems in turtle ears (e.g., *Evers & Benson, 2019*; *Evers et al., 2019*; *Foth et al., 2019*), and which is also present in *Uluops uluops* (UCM 53971). Because the fossa continues dorsolateral to the fenestra ovalis onto the opisthotic, we use the term perilymphatic fossa instead of posterior prootic recess as in some previous works (e.g., *Evers & Joyce, 2020*). Laterally to the perilymphatic fossa, the pterygoid and prootic form the posterior foramen for the canalis cavernosus. The mediodorsal wall of the canalis cavernosus, formed by the prootic, is incised by a sulcus for the hyomandibular nerve (Fig. 5B), which extends from the lateral foramen of the facial nerve canal to the posterior foramen for the canalis cavernosus. Such a sulcus is also present in *Uluops uluops* (UCM 53971).

**Opisthotic.** The opisthotics are nearly completely preserved on both sides of the cranium of UMZC T1041, but while the left one remains in articulation with the adjacent prootic and quadrate on the left side, the right one is ventrally displaced (Figs. 2B, 2F, 5C–5D). The opisthotic contacts the prootic anteriorly, the supraoccipital medially, the exoccipital posteroventromedially, the parabasisphenoid ventrally, the quadrate and squamosal laterally, and the pterygoid ventrally.

The anterior part of the opisthotic forms parts of the cavum labyrinthicum (Figs. 5C–5D). On the right opisthotic, the processus interfenestralis, which walls the cavum labyrinthicum posteriorly, is broken off (Fig. 5A). On the left element, the process is also damaged, but still preserved in articulation with the underlying pterygoid (Figs. 5C–5D). The processus interfenestralis is delicate in UMZC T1041, but resembles the process of *Eilaenchelys waldmanni* (NMS.G.2004.31.15) and more crownward positioned stem-turtles by reaching ventrally to fully separate the cavum labyrinthicum from the recessus scalae tympani. At the dorsal base of the processus interfenestralis, the lateral foramen for the glossopharyngeal nerve is visible (Figs. 5A, 5C). The ventral end of the process is expanded to a small footplate that articulates with the dorsal surface of the pterygoid, and frames the fenestra ovalis posteroventrally (Figs. 5C–5D). The footplate of the process also minimally contacts the parabasisphenoid, which has a dorsally high margin surrounding the 'cup' that holds parts of the brain. The processus interfenestralis is centrally broken on the left side of UMZC T1041, probably where the fenestra perilymphatica minimized its structural integrity. Whereas the ventral part with the footplate is in perfect articulation with the prootic and pterygoid, the dorsal part of the process and the remainder of the left opisthotic are slightly misplaced with regard to their original position. Although the fenestra perilymphatica is thus collapsed, there is no reason to believe its size was smaller than that of other turtles, as sometimes stated for paracryptodires and *Pleurosternon bullockii* specifically (e.g., *Brinkman & Nicholls, 1993*; *Gaffney, 1996*; *Joyce, 2007*). Additionally, the perilymphatic foramen of
*Uluops uluops* (UCM 53971) is of 'regular' size, and the morphology of the processus interfenestralis of this taxon is very similar to that of *Pleurosternon bullockii*. Laterally and slightly anteriorly to the base of the processus interfenestralis of UMZC T1041, there is a shallow sulcus in the roof of the cavum acustico-jugulare, the perilymphatic fossa, which continues anteriorly to become confluent with respective fossa on the prootic (Fig. 5C). The presence of this sulcus, its connection to the prootic fossa, and its posterior end at the base of the processus interfenestralis, and thus near the recessus scalae tympani, provide further tentative evidence that the fossa indeed is associated to the soft tissues that form an enclosed fluid-flow system between the fenestra ovalis, cavum labyrinthicum, foramen perilymphatica, and recessus scalae tympani (e.g., *Evers & Benson, 2019*; *Foth et al., 2019*).

A large facet for the exoccipital spans over the posteromedial margin of the opisthotic (Fig. 5A). The facet extends posteriorly halfway along the paroccipital process, indicating that the exoccipital had an elongate posterodorsolateral process. The size of the exoccipitals as inferred from their opisthotic facets suggests that the recessus scalae tympani, which is posteriorly framed by the exoccipital, was a large element. The dorsal margin of the foramen jugulare anterius is visible as a gentle notch in the opisthotic's margin with the brain cavity (Fig. 5A) and located just anterior to the facet for the exoccipital.

The paroccipital process of the opisthotic is relatively flat dorsoventrally (Figs. 2F, 5A), as in *Uluops uluops* (UCM 53971), and has a thin edge posterior to the articulation facet for the exoccipital. The lateral surface of the process lies against the quadrate and both are dorsally overlapped by the squamosal.

**Parabasisphenoid.** The parabasisphenoid of turtles develops from the fusion of the parasphenoid and basisphenoid (*Sterli et al., 2010*). Although some regin of the fused parabasisphenoid can be attributed relatively safely to the original ossification centers, as hypothesized by *Sterli et al. (2010)* for several taxa including *Pleurosternon bullockii*, the CT scan of UMZC T1041 gives no indication as to where the suture between both ossifications lies. Thus, the basisphenoid and parasphenoid are completely fused in UMZC T1041. In consequence, we segmented the parabasisphenoid as a single element, and describe it as such without distinguishing the parasphenoid and basisphenoid regions. The parabasisphenoid of UMZC T1041 is completely preserved and contacts the pterygoid, prootic, opisthotic, and basioccipital (Figs. 2B, 7). A contact with the exoccipital is not preserved, but cannot be ruled out, as well as possible anterior contacts with the palatines or the vomer. The parabasisphenoid of *Pleurosternon bullockii* is anteroposteriorly elongate and mediolaterally narrow (Fig. 7). The ventrally exposed surface of the parabasisphenoid is vaguely triangular, as the parabasisphenoid narrows anteriorly to the rostrum basisphenoidale. Although the pterygoids are not in perfect articulation with the parabasisphenoid in UMZC T1041, it seems that the ventral surface of the rostrum basisphenoidale was exposed ventrally, separating the pterygoids and preventing them from having a midline contact (see also *Evans & Kemp, 1975*). Indication for this interpretation comes from anteroposterior grooves to either side of the rostrum basisphenoidale (Fig. 7B), which we interpret to be the articular facets of the pterygoid. The ventral surface of the parabasisphenoid between these facets is completely smooth, just as the adjacent ventral surface of the pterygoids, further support

the notion that this bone was exposed. Conversely, parts of the parabasisphenoid that are overlapped by the pterygoid are usually roughened by articular ridges in comparative taxa like *Uluops uluops* (UCM 53971). A ventrally exposed rostrum basisphenoidale has also been described for *Glyptops ornatus* (*Gaffney, 1979a*), but it remains unclear, though plausible, if the pterygoids are fully separated by the parabasisphenoid in both taxa.

At about half the length of the parabasisphenoid of UMZC T1041, the basipterygoid process projects laterally from the ventrolateral margin of the bone (Figs. 7B–7C, 7E). Although it has been argued that the process on the lateral margin of the basisphenoid of UMZC T1041 does not constitute a basipterygoid process (*Gaffney, 1979a*; *Sterli et al., 2010*), we agree with the original identification of *Evans & Kemp (1975)* and later assessments of the structure (e.g., *Rabi et al., 2013*). The basipterygoid process is subdivided into a dorsal and a ventral flange by a deep sulcus for the cerebral artery (Fig. 7E), which is also the case in *Uluops uluops* (UCM 53971). The dorsal flange is likely homologous with the basipterygoid process of other turtles, as the carotid system extends ventrally to it. The cerebral artery canal of UMZC T1041 fully enters the parabasisphenoid just anterior to the basipterygoid process via the foramen posterius canalis carotici cerebralis (see also *Sterli et al., 2010*) and projects anteromedially through the bone (Fig. 7E). The dorsal flange of the basipterygoid process is larger, and inserts deeply into the respective facet on the pterygoid. Anterior to the foramen posterius canalis carotici cerebralis, there is a faint and narrow groove that extends along the ventrolateral surface of the rostrum basisphenoidale. This groove is herein not interpreted as a narrow canal for the palatine artery, as it is not mirrored in the pterygoid bone. Instead, the pterygoid would fill the groove when articulated tightly with the parabasisphenoid. Thus, we herein interpret the groove as the delimitation of the facet for the pterygoid (Figs. 7B, 7E). The palatine canal therefore appears to be absent in UMZC T1041.

A peculiar feature of the basicranium of UMZC T1041 is the presence of a posterior process jointly formed by the parabasisphenoid and pterygoid (Figs. 7B–7C, 7E), which overlaps the ventral surface of the basioccipital. The parabasisphenoid part of this process is larger and more distinct than the pterygoid part. In UMZC T1041, the ventral surface of the posterior parabasisphenoid processes is slightly raised, and forms a shallowly concave fossa between them. Additionally, the surface of the process is textured by a short ridge. Such parabasisphenoid processes are also present in *Glyptops ornatus* (*Gaffney, 1979a*) and *Uluops uluops* (UCM 53971), and could be a pleurosternid synapomorphy. Posterolaterally projecting processes of the parabasisphenoid are also present in some baenids, such as *Eubaena cephalica* (*Rollot, Lyson & Joyce, 2018*), but the baenid process has less external relief, and seems better integrated into a flat ventral surface of the basicranium. Similar structures are also found in helochelydrids, in which these are sometimes referred to as a secondarily pair of tubercula basioccipitale (*Joyce, Sterli & Chapman, 2014*). In helochelydrids, these anterior tubercula basioccipitale are predominantly formed by the pterygoids (*Naomichelys speciosa*: *Joyce, Sterli & Chapman, 2014*), or entirely formed by the pterygoids (*Helochelydra nopscai*: *Joyce et al., 2011*). Despite these differences in composition of the processes, the anterior tubercula basioccipitale of helochelydrids and the posterior parabasisphenoid processes of pleurosternids share (i) that the posterior

margin between them is concave; (ii) that they are slightly raised and form a concavity medially between them; (iii) that they overlap the basioccipital; and (iv) that their surface is textured indicating soft tissue insertion. Thus, topological and anatomical arguments suggest the homology of the pleurosternid and helochelydrid processes, and possibly provide evidence for paracryptodiran affinities of helochelydrids (see also *Joyce, 2017*; *Joyce & Anquetin, 2019*). To highlight this possible homology, we follow *Joyce, Sterli & Chapman (2014)* by calling these processes (anterior) tubercula basioccipitale (Fig. 7).

The dorsal surface of the parabasisphenoid of UMZC T1041 is extremely similar to the same surface in *Uluops uluops* (UCM 53971) and YPM 4717, a specimen referred to *Glyptops plicatulus* by *Gaffney (1979a)* (*Glyptops ornatus* according to taxonomy followed here) based on temporal and geographic reasoning. It can be divided into a posterior, slightly cup-like region that holds parts of the brain, and the rostrum basisphenoidale anteriorly (Fig. 7A). In the posterior region, the parabasisphenoid is dorsoventrally thickest. The posterolateral margin of the parabasisphenoid is dorsally expanded. This expansion, which is also present in *Uluops uluops* (UCM 53971), forms the ventral margin of the hiatus acusticus (Fig. 7A) and contacts the processus interfenestralis of the opisthotic posteriorly and the prootic anteriorly (Fig. 5D). The lateral margins of the parabasisphenoid anterior to the hiatus acusticus margin are gently raised, so that the medial space between is transversely concave (Fig. 7A). The concavity is interrupted posteriorly by a shallow basis tuberculi basalis (Figs. 7A, 7D), which would presumably have continued posteriorly onto the basioccipital, as seen in *Uluops uluops* (UCM 53971). Foramina for the abducens nerve are present anterolaterally on the dorsal parabasisphenoid surface. The respective anterior foramina exit on the anterior basisphenoid surface within the retractor bulbi pits ventral to the clinoid processes. The clinoid processes at the anterolateral corner of the parabasisphenoid cup are extremely short (Figs. 7A, 7D), as is also the case in *Uluops uluops* (UCM 53971). The dorsum sellae between the clinoid processes of UMZC T1041 is not developed as a vertical wall, but also not as a horizontal sheet that overlaps the sella turcica. Instead, the dorsum sellae is only a very minor, shallow ridge (*Evans & Kemp, 1975*; Fig. 7D). Ventrally underneath the ridge, the anterior surface of the parabasisphenoid slopes toward the rostrum basisphenoidale and sella turcica. This anterior surface is gently depressed and a midline ridge is absent (Fig. 7D). The lateral aspects of the anterior surface are delimited by short vertical ridges that extend from the base of the clinoid process downwards. These ridges also define deep retractor bulbi pits directly ventral to the clinoid processes (Fig. 7D). The retractor bulbi pits are further accentuated dorsally by a horizontal ridge at the base of the clinoid process, which is the same in *Glyptops ornatus* (*Gaffney, 1979a*). In *Uluops uluops* (UCM 53971), the anterior surface of the parabasisphenoid varies from *Glyptops ornatus* and *Pleurosternon bullockii* in lacking distinct retractor bulbi pits and having a deeper anterior surface ventral to the dorsum sellae. The rostrum basisphenoidale of UMZC T1041 extends anteriorly as a mediolaterally narrow sheet of bone. Its lateral margins are dorsally upturned, so that the sella turcica is well defined as a deep fossa (Figs.

7A, 7D). In the posterior margin of the sella turcica, the paired foramina anterius canalis carotici cerebralis are situated in relatively close proximity to one another (Fig. 7D).

**Stapes.** The stapes are not preserved for UMZC T1041.

**Carotid circulation and facial nerve.** Unambiguous osteological correlates are present in UMZC T1041 for the cerebral artery, the (undivided) facial nerve, and both major distal branches of the facial nerve, which are the hyomandibular and vidian nerves. As there is no evidence for a palatine artery canal, the palatine artery is either absent, or uncovered by bone, but located in an unusual position.

As described above (see parabasisphenoid), the basipterygoid process of UMZC T1041 is subdivided into a dorsal and a ventral flange (Figs. 7B–7C, 7E), between which the canal for the cerebral artery starts to pierce the parabasisphenoid (Fig. 7E). This region is ventrally exposed, as the parabasisphenoid and pterygoid form a window-like recess around the basipterygoid articulation. No canalis caroticus internus is present posterior to the position of the basipterygoid process, so that the internal carotid artery is not embedded in bone (see also *Sterli et al., 2010*). As the ventral flange of the basipterygoid process is much smaller than the dorsal flange, the artery can enter the basicranium via the small fenestra between the pterygoid and parabasisphenoid. As the artery directly enters the parabasisphenoid in this position, we do not classify this opening within the basipterygoid articulation as a foramen posterius canalis carotici interni (as done in *Evans & Kemp, 1975*), but as the foramen posterius canalis carotici cerebralis (as in *Sterli, 2010*). The palatine artery canal, if present, would be expected to extend anteriorly between the pterygoid and parabasisphenoid from the position of the foramen posterius canalis carotici cerebralis. *Gaffney (1979a)* indicated the presence of such a canal for *Glyptops ornatus*, but this should be revisited, as he did not find the anterior foramen for the palatine artery canal and thus might have misidentified the foramen for the vidian canal. In UMZC T1041, no palatine artery canal can be discerned. The parabasisphenoid, which is slightly disarticulated from the left pterygoid and strongly so from the right one, shows a subtle and narrow groove, that extends from the foramen posterius canalis carotici cerebralis forward and could thus be a candidate structure for the palatine artery canal (Fig. 7B). However, when the pterygoid is articulated, the respective groove is 'closed' by bone of the pterygoid, so that the groove likely constitutes a pterygoid facet, rather than a tiny canalis caroticus palatinum. As the posterior course of the internal carotid artery prior to its 'split' is not embedded in bone, the absence of the palatine artery canal does not necessarily warrant concluding that the artery itself is absent. However, as there is no indication of an interpterygoid slit or similar opening through which the palatine artery could eventually enter the cranium in UMZC T1041 or related taxa with more completely preserved anterior palate regions, the most parsimonious explanation of the observed pattern indeed is the loss of the palatine artery. The loss of the palatine artery has also been inferred for other paracryptodires (*Gaffney, 1975a*; *Lipka et al., 2006*), including baenids (*Rollot, Lyson & Joyce, 2018*). However, the absence does not seem universal, as *Uluops uluops* clearly shows a palatine artery canal (UCM 53971). A palatine artery canal has also been reported for *Dorsetochelys typocardium* by *Anquetin & André (2020)*, but these authors do not describe the canal system in detail

and it thus remains unclear if the proposed palatine artery canal foramen reported for *Dorsetochelys typocardium* is indeed that or pertains to the canal system of the vidian nerve. Indeed, the posterior entry of the palatine artery described by *Anquetin & André (2020)* resembles that of the vidian canal described herein for *Pleurosternon bullockii* by piercing the pterygoid. As the situation in *Dorsetochelys typocardium* demands clarification, ideally by means of CT scanning, we will not further comment on this taxon herein. The condition of *Uluops uluops* (UCM 53971) is very similar to that *Pleurosternon bullockii* by having the carotid canal enter within a bifurcated basipterygoid process, although the former also has a foramen for the palatine artery. *Uluops uluops* (UCM 53971) shows, that no part of its internal carotid artery up to the point of its division is embedded by bone. In *Arundelemys dardeni*, the internal carotid artery itself is also not encased anteriorly, the palatine artery is absent, and the entry of the cerebral artery canal lies within a shallow fossa between the parabasisphenoid and pterygoid. All these turtles differ from baenids, in which the anterior part of the internal carotid artery is concealed by bone. This embedding of the internal carotid artery in baenids extends posteriorly up to the intersection of the carotid artery system with the canalis pro ramo nervi vidiani (*Rollot, Lyson & Joyce, 2018*). On the other hand, similarities between baenids, *Pleurosternon bullockii*, and *Arundelemys dardeni* exist in their shared absence of a palatine artery. Thus, most paracryptodires show similarities to one another in their carotid artery system, but lots of within-group variation exists that can probably only be better understood once the phylogenetic relations of these turtles are better characterized.

The facial nerve of UMZC T1041 exits the braincase laterally through the canalis nervus facialis, which extends from the fossa acustico-facialis through the prootic and into the canalis cavernosus. The position of the geniculate ganglion can be inferred to be within the canalis cavernosus for UMZC T1041, because direct osteological evidence for the course of the hyomandibular and vidian nerves begin there: the prootic has a hyomandibular sulcus that extends from the lateral facial nerve canal foramen posteriorly to the cavum acustico-jugulare (Fig. 7B). The vidian nerve passes ventrally through the pterygoid via the canalis pro ramo nervi vidiani, which exits just posterior to the pterygoid ridge that borders the area of the basipterygoid articulation (Fig. 7C). In turtles with an embedding of the internal carotid artery, the canalis pro ramo nervi vidiani usually leads from the canalis cavernosus into the internal carotid canal (e.g., *Gaffney, 1979b*). In UMZC T1041, which lacks an internal carotid canal, the vidian nerve is thus transmitted to the ventral surface of the basicranium. From here, it is inferred to pass between the dorsal and ventral flanges of the basipterygoid process, together with the carotid artery. Within the facet for the basipterygoid process, the vidian nerve then enters the canalis nervus vidianus, which starts in the center of the facet and extends anteriorly through the pterygoid. This posterior foramen for the vidian canal was interpreted as a foramen posterius canalis carotici palatinum by *Sterli et al. (2010)*. However, the canal extends through the pterygoid only, instead of passing along the parabasisphenoid-pterygoid suture, and exits anterolaterally from the sulcus cavernosus (Figs. 6B, 6C), instead within the sulcus itself. Thus, the course and exit foramen of the canal conform to the expectations of a canalis nervus vidianus (*Rollot, Lyson & Joyce, 2018*).

## DISCUSSION

**Paracryptodiran monophyly.** The fossil turtle clade Paracryptodira was initially proposed based on the anterior position of the foramen posterius canalis carotici interni halfway along the contact of the basisphenoid with the pterygoid, in contrast to pleurodires, which have an entry that involved the prootic and/or quadrate, and crown cryptodires, which have an entry towards the posterior margin of the pterygoid (*Gaffney, 1975a*). The accuracy and utility of this character has been debated ever since (e.g., *Evans & Kemp, 1976*; *Gaffney, 1979a*; *Rieppel, 1980*; *Sterli et al., 2010*; *Rabi et al., 2013*), but progress has been hampered by confusing terminology and conflicting observations.

Our study suggests that two distinct morphotypes are present among paracryptodires. Whereas an anteriorly positioned foramen posterius canalis carotici interni indeed seems to be present in *Compsemys victa* (UCM 53971) and *Eubaena cephalica* (*Rollot, Lyson & Joyce, 2018*), this foramen is not present in *Pleurosternon bullockii* (UMZC T1041) and *Uluops uluops* (UCM 53971), as the internal carotid artery is not embedded in bone in these taxa. This difference was initially disguised by terminology used from the 1970s to early 2000s, as the foramen posterius canalis carotici cerebralis was addressed as the foramen posterius canalis carotici interni (*Rabi et al., 2013*). Homology can nevertheless be maintained, if the foramen posterius canalis carotici interni is conceptualized as the entry to a small pit at the basisphenoid/pterygoid suture from which the cerebral and palatine arteries penetrate the surrounding bones, as had been reported for several paracryptodires, including *Glyptops ornatus* (Gaffney, 1979) or *Pleurosternon bullockii* (*Sterli et al., 2010*), but we are here able to demonstrate that this character concept is at least inapplicable to the latter, as the palatine canal is absent.

The above-listed differences should not be used to dismiss the fact that the carotid system, in general, enters the skull in all known paracryptodires from below relatively far anteriorly at or near the basisphenoid/pterygoid suture. Indeed, as the exact vessels penetrating the skull could only rarely be assessed prior to the use of CT scanning technology, we suspect from personal experience (WGJ) that the paracryptodiran condition was conceptualized by most authors by reference to topology, not the exact vessels that enter the skull. This broadened character concept, however, still pertains to two unrelated morphological aspects: first, the anterior placement of the entry of the carotid system and, second, the presence of an extended basisphenoid-pterygoid contact. The (derived) presence and extent of the basisphenoid-pterygoid contact is addressed in current phylogenetic analyses by other characters (e.g., *Joyce, 2007*; *Sterli, 2010*; *Anquetin, 2012*; *Zhou & Rabi, 2015*; *Evers & Benson, 2019*), while the (plesiomorphic) anterior placement of the carotid system entry could be grouped with the condition seen in basal turtles. So, while these two morphological aspects combined indeed diagnose paracryptodires relative to most other turtles, they are redundant with existing characters. We therefore here further support recent phylogenetic analysis (e.g., *Evers & Benson, 2019*) by suggesting that the 'paracryptodiran condition' should not be utilized as a carotid arterial character.

Although the most important 'paracryptodiran' cranial character can thus be debunked, pleurosternids, baenids, and other potential paracryptodires share several features that

could potentially support their monophyly. In addition to the sculpturing of the shell (present in early baenids; e.g., *Gaffney, 1972*; *Joyce & Anquetin, 2019*), baenids and pleurosternids share some gross resemblance in their maxilla-premaxilla morphology, including a relatively strongly curved labial margin of the maxilla and a notch in the median section of the labial ridge along the premaxilla (*Gaffney, 1972*). Additionally, the jugal is dorsally removed from posteriorly continuing the labial ridge otherwise formed by the maxilla (*Gaffney, 1972*). These features are, however, not unique to paracryptodires, as they also appear in extant turtles with ecologies that are probably similar to those of paracryptodires, such as riverine geoemydids (e.g., *Gaffney, 1979b*). To critically assess potential new paracryptodiran synapomorphies, the skulls of early baenids, particularly *Trinitichelys hiatti*, should be re-described.

**Paracryptodiran relationships.** This contribution is part of a larger study that attempts to better resolve the internal and external relationships of paracryptodires, if monophyletic, through a better understanding of their anatomy. Although the novel observations we made herein will be processed in later contributions, we take the liberty of highlighting similarities that particularly intrigue us and that may have phylogenetic significance.

Irrespective of paracryptodiran monophyly, our description of *Pleurosternon bullockii* provides morphological evidence for the content of Pleurosternidae, based on instances of similarities of *Pleurosternon bullockii* with some other taxa. Although a re-description of *Glyptops ornatus* is warranted to fully appreciate its anatomy, it is clear from existing information (e.g., *Gaffney, 1979a*) that this taxon is indeed very similar to *Pleurosternon bullockii*, both in terms of general skull form, and in more detailed anatomical aspects. For instance, *Pleurosternon bullockii* and *Glyptops ornatus* share the following features: small prefrontals; long anterior processes of the frontals that partially separated the nasals; a squamosal-parietal contact; dorsal position of the jugal above the level of the labial margin of the maxilla; and, among other similarities in the parabasisphenoid morphology, the presence of anterior tubercula basioccipitale, the presence of a basipterygoid process, and the presence of deep retractor bulbi pits. The same is generally true for *Dorsetochelys typocardium*, in particular the jugal exclusion from the orbit (DORCM G.00023; *contra Evans & Kemp, 1976*) and the presence of anterior tubercula basioccipitale (DORCM G.00023; not discernible in *Evans & Kemp, 1976*). In both cases, these similarities are not surprising, given that more recent phylogenetic hypotheses already highlight close relationships among these turtles (e.g., *Pérez-García, Royo-Torres & Cobos, 2015*; *Joyce & Rollot, 2020*).

Although the skull of *Uluops uluops* deviates in general aspects of its cranial shape, we note a great number of similarities between this taxon and *Pleurosternon bullockii*. Although *Uluops uluops* has so far not been considered a pleurosternid (e.g., *Carpenter & Bakker, 1990*; *Lyson & Joyce, 2011*; *Pérez-García, Royo-Torres & Cobos, 2015*; *Joyce & Rollot, 2020*), our comparisons indicate that it might be attributable to this clade. Among the features that are shared between *Uluops uluops* and *Pleurosternon bullockii* are a squamosal-parietal contact (possibly plesiomorphically present in baenids; *Gaffney, 1972*); the exclusion of the prootic from the trigeminal foramen by a posteroventral ramus of the parietal (also

present in thalassochelydians and sandownids; *Anquetin, Püntener & Joyce, 2017*; *Evers & Joyce, 2020*); the presence of an extended sulcus palatino-pterygoideus formed in part by an enlarged septum orbitotemporale (similar to pleurodires, but also present in some cryptodires; *Gaffney, Tong & Meylan, 2006*); presence of a deep fossa in the roof of the sulcus palatino-pterygoideus (also in pleurodires; *Gaffney, Tong & Meylan, 2006*); dorsal position of the jugal above the level of the labial margin of the maxilla (also present in baenids; *Gaffney, 1972*); the size and shape of the quadratojugal; the presence of a small quadrate-supraoccipital contact (unclear in early baenids); the complete surrounding of the fenestra ovalis by the prootic and opisthotic (absent in *Arundelemys dardeni* [USNM 41614]; unclear in early baenids); the presence of a hyomandibular nerve sulcus in the posterior part of the canalis cavernosus (unclear in early baenids); the presence of anterior tubercula basioccipitale (shared also with helochelydrids, see below); the presence of a basipterygoid process that is laterally invaded by the cerebral artery (absent in baenids and *Compsemys victa*, but symplesiomorphic for turtles); and the general shape of the parabasisphenoid (see description for full details). Although some of these characters may prove to be more widespread among paracryptodires in particular or stem-turtles more generally, the number of similar features between *Pleurosternon bullockii* and *Uluops uluops* lend support to the hypothesis that the latter is a pleurosternid.

We here also document intriguing similarities between *Pleurosternon bullockii* and helochelydrids, particularly in the anterior tubercula basioccipitale of the parabasisphenoid, but also in other features, such as the exclusion of the jugal from the orbit (e.g., *Joyce et al., 2011*; *Joyce, Sterli & Chapman, 2014*; *Joyce, 2017*). Helochelydrids should be integrated into phylogenies with a dense sampling of paracryptodires to test if they are more closely related than currently thought.

We also note the presence of some features in *Pleurosternon bullockii* that are possibly more widespread among crownward stem-turtles than previously recognized, and which can therefore possibly aid in better constraining the global position of pleurosternids. For instance, a pterygoid-basioccipital contact has previously not universally been recognized in pleurosternids (but see *Gaffney, 1979a*). This feature, considered diagnostic for baenids (*Joyce & Lyson, 2015*), is present in *Pleurosternon bullockii* and *Uluops uluops* (as well as in compsemydids and helochelydrids). Although thus probably widely present among paracryptodires, this feature probably cannot serve as a paracryptodiran synapomorphy, as several other clades that have repeatedly been hypothesized to be crownward stem turtles (e.g., *Joyce, 2007*), including thalassochelydians, protostegids, and some xinjiangchelyids, also possess a pterygoid-basioccipital contact (*Brinkman et al., 2013*; *Anquetin, Püntener & Joyce, 2017*; *Evers, Barrett & Benson, 2019*). Conversely, we here demonstrate that a purported thalassochelydian synapomorphy, the presence of a posteroventral process of the parietal along the posterior margin of the trigeminal foramen (*Anquetin, Püntener & Joyce, 2017*), is also present in *Pleurosternon bullockii* (and also *Uluops uluops*). The two examples of the pterygoid-basioccipital contact and the presence of a posteroventral parietal process are a nice demonstration of how alleged synapomorphies or diagnostic features of relatively inclusive groups can have a much wider distribution across phylogeny. This highlights (i) the importance of detailed descriptive work to acknowledge the anatomical

variation present across several clades; and (ii) the importance of global phylogenetic approaches to the classification of fossil-only turtle clades. The latter is because the character state distribution outside the 'focal clade' has importance not only for inference of global phylogenetic placements, but also has an effect on in-group relationships and the optimization and polarity of characters (e.g., *Kluge & Farris, 1969*; *Farris, 1972*; *Nixon & Carpenter, 1993*). The distribution of the pterygoid-basioccipital contact, for instance, indicates that rather than being a local baenid synapomorphy, it possibly represents a synapomorphy of the clade that includes paracryptodires but also other probable crownward stem-turtle clades (e.g., *Evers & Benson, 2019*) such as xinjiangchelyids or thalassochelydians. On the other hand, studies focused on more inclusive focal groups without a global phylogenetic perspective often sample characters that document variation among the suspected clade members to a finer level of detail than done in most global studies (e.g., Joyce & Lyson, 2011, for paracryptodires). We think that it is important to reconcile the finer-level studies with global phylogenetic approaches to address questions such as: do baenids and paracryptodires form a monophylum?; or what is the global position of these clades? We hope that detailed anatomical studies such as the one presented here lead to the recognition of new phylogenetic characters as well as the revision of those that have been used in the past, so that the outlined questions can be more comprehensively addressed in the future.

On a final note, we find that *Pleurosternon bullockii* shows some remarkable similarities with pleurodire turtles, particularly in the orbitotemporal region of the skull, as well as the external process of the pterygoid. However, the orbitotemporal region of most fossil turtles is not sufficiently described to draw meaningful comparisons from the literature alone. CT scans of several specimens available to us, such as the xinjiangchelyid *Annemys* sp. (IVPP V18106; specimen described in *Brinkman et al. (2013)*), the probable sandownid *Solnhofia parsonsi* (TM 4023; specimen described in *Gaffney (1975b)*) or the plesiochelyid *Plesiochelys planiceps* (OUMNH J1582; specimen described in *Gaffney (1976)*) show that a strong ventral ridge of the postorbital that would form the septum orbitotemporale and define a narrow sulcus palatino-pterygoideus is absent in these turtles. This indicates that the similarity observed between *Pleurosternon bullockii* and pleurodires in regard to the septum interorbitale and related sulcus palatino-pterygoideus is unique, and not more widespread across the turtle stem. Similarly, the pleurodire-reminiscent morphology of the external pterygoid process of *Pleurosternon bullockii* is not seen in other crownward stem-turtles: the vertical flange of the external process of the pterygoid of *Pleurosternon bullockii* is comparatively larger than that of plesiochelyids or *Annemys* sp. Although both the size of the vertical flange and its slight medial deflection at the dorsal margin in *Pleurosternon bullockii* establishes some similarity to the pterygoid trochlea of pleurodires, important differences remain. For instance, the surface of the vertical flange of the external pterygoid process is oriented parallel to the sagittal skull plane, and not posterolaterally inclined. Unfortunately, the morphology of the pterygoid trochlea is unknown for stem-pleurodires (*De Lapparent de Broin, de la Fuente & Fernandez, 2007*), but a gradual evolution of the pterygoid adductor musculature system via a double-trochlea along the external pterygoid process and the otic capsule has been proposed as a possible

evolutionary scenario (e.g., *Joyce, 2007*). Although the presence of a double trochlea does not seem to provide biomechanical advantages according to analytical studies (*Ferreira et al., 2020*), all hypotheses for the evolution of the pleurodire trochlea invoke a size increase of the external pterygoid process and its lateral surface as an initial step (e.g., *Joyce, 2007*; *Joyce & Sterli, 2012*; *Ferreira et al., 2020*). Although, at this point in time, we do not hypothesize that pleurosternids are stem-pleurodires, the similarities noted herein should be further explored by additional comparisons and phylogenetic implementation.

## CONCLUSIONS

The cranial anatomy of *Pleurosternon bullockii* and comparative descriptions confirm high levels of similarity with other hypothesized pleurosternids, such as *Glyptops ornatus* or *Dorsetochelys typocardium*. Additionally, we note a large number of similarities between *Pleurosternon bullockii* and *Uluops uluops*, which possibly is a pleurosternid as well. Among other features, these taxa share a similar basipterygoid region, with the symplesiomorphic retention of a basipterygoid process and a ventrally unimbedded internal carotid artery course, shared with earlier diverging stem-turtles. These features, along with a relatively extensive dorsal skull coverage and weakly developed otic trochleae, support the hypothesis that pleurosternids are globally positioned among an extended stem-lineage of turtles. Our anatomical description of *Pleurosternon bullockii* provides tentative evidence for close relationships between pleurosternids and helochelydrids, which share unusual posterior processes of the parabasisphenoid that form a second set of (anterior) tuberculae basioccipitale.

Although variation regarding the internal carotid region of the cranium exists among pleurosternids, it is clear that the anterior position of the foramen posterius canalis carotici interni, historically used as a synapomorphy to unite baenids and pleurosternids as Paracryptodira, is not present in pleurosternids, and results from misinterpretation of the respective anatomy. As other paracryptodiran cranial synapomorphies are also dubious, our observations challenge previous observations upon which traditional hypotheses regarding the systematics of paracryptodires were built. Although the monophyly of paracryptodires is thus currently not supported by evidence from cranial anatomy, CT documentation of early baenid cranial morphology is outstanding, and may provide such evidence in the future.

*Pleurosternon bullockii* shows a surprising number of similarities with pleurodires, which are seen in the area between the otic and temporal fossae. The formation of a relatively narrow sulcus palatino-pterygoideus by an expanded septum orbitotemporale is unparalleled in other crownward stem-turtles such as xinjiangchelyids or thalassochelydians. An expanded external pterygoid process with a dorsomedially deflected vertical flange also approaches the pleurodiran morphology. Further investigations are necessary to test if this morphology could be a model for an ancestral pleurodiran pterygoid trochlea.

**Institutional Abbreviation**

**DORCM**    Dorset County Museum, Dorchester, United Kingdom

| IVPP | Institute of Vertebrate Paleontology and Paleoanthropology, Chinese Academy of Sciences, Beijing, People's Republic of China |
|------|------|
| NMS | National Museum of Scotland, Edinburgh, United Kingdom |
| OUMNH | Oxford University Museum of Natural History, University of Oxford, Oxford, United Kingdom |
| UCM | Museum of Natural History, University of Colorado, Boulder, Colorado, USA |
| UMZC | Museum of Zoology, Cambridge University, Cambridge, United Kingdom |
| USNM | United States National Museum, Smithsonian Institution National Museum of Natural History, Washington, D.C., USA |
| TM | Teylers Museum, Haarlem, The Netherlands |
| YPM | Yale Peabody Museum of Natural History, New Haven, Connecticut, USA |

## ACKNOWLEDGEMENTS

We thank Roger Benson (University of Oxford) for scanning the cranium of UMZC T1041 in our stead. Keturah Smithson is thanked for maintaining the scanner at the Cambridge Biotomography Center and Jason Head (University of Cambridge) is acknowledged for permission to study the specimen. We also thank Gabriel Ferreira, Olivier Rieppel, and Julien Claude for three reviews that helped to improve a previous version of this manuscript, and we thank Fabien Knoll for the editorial oversight of our submission. Particular thanks to Gabriel Ferreira for spotting a segmentation mistake that would have been annoying if it had been published.

### Funding

The work is supported by a grant from the Swiss National Science Foundation (SNF 200021_178780/1). The funders had no role in study design, data collection and analysis, decision to publish, or preparation of the manuscript.

### Grant Disclosures

The following grant information was disclosed by the authors:
Swiss National Science Foundation: SNF 200021_178780/1.

### Competing Interests

The authors declare there are no competing interests.

### Author Contributions

- Serjoscha W. Evers conceived and designed the experiments, performed the experiments, analyzed the data, prepared figures and/or tables, authored or reviewed drafts of the paper, and approved the final draft.
- Yann Rollot and Walter G. Joyce conceived and designed the experiments, prepared figures and/or tables, authored or reviewed drafts of the paper, and approved the final draft.

## Data Availability

The CT data of the cranium of Pleurosternon bullockii (UMZC T1041) and 3D models resulting from the segmentation of this dataset are available at MorphoSource (Project ID 1001, specimen media ID M63373).

https://www.morphosource.org/Detail/ProjectDetail/Show/project_id/1001.

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
