# Peer review of "Cranial osteology of the Early Cretaceous turtle *Pleurosternon bullockii* (Paracryptodira: Pleurosternidae)"

_PeerJ, doi:10.7717/peerj.9454_

## Round 0.1 · original submission · Minor Revisions

The reviewers are very positive about your work, but still offer some suggestions for improvements. Please, check in particular that the caudal fragment of the left premaxilla is genuine. I agree with Reviewer 1 that it may well be in fact a fragment of the maxilla.

As per PeerJ policies (https://peerj.com/about/policies-and-procedures/#data-materials-sharing), all the raw data have to be made available in a permanent public repository prior to formal acceptance.

Please, together with your unmarked revised manuscript, provide a marked-up copy as well as a document explaining how you have addressed the points raised by the reviewers.

·

Basic reporting

This manuscript by Evers, Rollot and Joyce is an outstanding contribution to comparative anatomy of turtles. The authors thoroughly redescribe Pleurosternon bullockii based on a CT-scan of a single well-preserved and partially articulated specimen and compare their findings to a good sample of other paracryptodires and the condition of other turtles, such as helochelydrids and pleurodires. The manuscript is well-written in perfect English and the discussion is well-supported by the data. The introduction covers the relevant context and the figures are beautifully present, and adequately illustrated the discussed topics. I have only some minor comments and suggestions which I provide on the attached commented PDF file. One of these needs to be addressed because it might be an error in the 3D segmentation of the described specimen. As such, I fully support the publication of this manuscript after this single issue is addressed by the authors.
The authors are welcomed to know my identity.

Experimental design

The authors use 3D reconstructions and renderings based on a micro-CT scan of a single specimen to build their redescription. All CT scan parameters are presented and the methods are well outlined.

Validity of the findings

The manuscript contains several important and novel anatomical findings and reinterpretations that will surely serve as the basis for novel phylogenetic interpretations of paracryptodires and turtles in general. I particularly enjoyed reading the comparisons between the described specimens and pleurodires, and I believe they will be very important for discussions in the turtle community. As such, this contribution will be of interest for experts in turtle anatomy and evolution.

Additional comments

As explained above, I have only some minor comments that are all available on the PDF file. It is worth to highlight just three of the most important ones.
(1) Between lines 276 and 279 of the manuscript PDF where you discuss the position of the jugal in the ventrolateral margin of the skull. You state that it forms a step-like margin between the maxilla and jugal, which is ok for your specimen, but you cite Glyptops as also possessing this feature. Although I've never seen a Glyptops specimen in person, based on its original description publication I find its margin very similar to that of Dorsetochelys, and not step-like. I might have not understood what you meant by a "step-like margin". Please, double-check this feature.
(2) You alternately used "cheek" and "lower temporal" emargination along the text. Please, choose one and be consistent all along it.
(3) The right and left premaxilla reconstructions seem conflicting to me. The left premaxilla is much larger, including an ascending process that forms part of the lateral rim of the external nare, which is shown on the right side as completely formed by the maxilla. Also, the anterior extension of the maxilla on the right side reaches the smaller premaxilla, which is compatible with the most anterior piece of the left premaxilla. This raises me the question of whether the second piece of the left premaxilla isn't, in fact, a piece of the maxilla instead, in which a crack was confused with a sutural contact. In fact, in all of the other taxa in your comparison sample, e.g. Compsemys, Dorsetochelys, Glyptops, and Arundelemys, the premaxilla does not extend so far posteriorly, which agrees with my interpretation above and also to the size and position of the right premaxilla. It would be very important for you to double-check this, so as to avoid misinterpreting its anatomy.

·

Basic reporting

no comment

Experimental design

no comment

Validity of the findings

no comment

Additional comments

I enjoyed reading this paper for several reasons: it is well-written (the occasional typo should be easy to correct using ‘spell-check’), exhaustingly referenced, and beautifully illustrated. The paper presents a detailed morphological description and analysis of a fossil turtle skull (Pleurosternon bullockii) from the Early Cretaceous of Europe, based on CT scans. The skeletal structures and associated soft anatomy (cranial nerves and blood vessels, jaw adductor musculature) is competently described in detail and exhaustively compared with a diversity of potentially closely related taxa. I am no expert in fine-grained turtle interrelationships, so I cannot comment critically on the choice of taxa employed in these comparisons. But the pleurodiran features shared by Pleurosternon bullockii are certainly particularly intriguing.
I think the authors do a particularly good job at a detailed analysis of cranial morphology in a suite of taxa that are compared with Pleurosternon bullockii, carefully evaluating the potential to derive from those descriptions and comparisons phylogenetically informative characters, without rushing ahead to present another global analysis of turtle interrelationships. The paper is exemplary for how to lay a solid partial groundwork for future such analyses. It is refreshing to see the authors concentrate on morphological details, rather than on character coding. I recommend the paper to be accepted for publication once the occasional typos have been cleaned up.

·

Basic reporting

It is a good and well written paper that will be useful for the community interested in turtle morphological evolution. I have a series of small remarks that are listed in "general comments".

Experimental design

no comment

Validity of the findings

no comment

Additional comments

This is an interesting study documenting the cranial anatomy of Pleurosternon bullocki. The paper is good, and my remarks concerns minor things that may help authors to tune the paper.

1. The authors argues that it is important for phylogenetic reconstruction, but they did not re-evaluate phylogenetic reconstructions with the characters they found and the similarity for Pleurodira. This is not a problem, but I think that these arguments could be likely dropped from the abstract and introduction since indeed better documenting morphology always help to better document phylogenies (but not only, they can also help to better understand the evolution of morphological functions). We have here a good redescripition of the skull of this species and there is no need to present that need as a solution for resolving a systematic problem since the paper is not demonstrating that we know better understand the relationship of paracryptodira and within that group. I am, however, fine with the discussion about character importance.

2. you often do some general comparisons to cryptodira and pleurodira without giving example of taxa; it should be good from the beginning if your comparison extend to cryptodira sensu stricto, or to pan cryptodira (and the same for pleurodira). You also do some comparison with Mesochelydia, maybe you should refer to Joyce 2017, when you do it.

3. Can the authors appreciate the size of the foramen stapedio-temporale and compare it with trionychids and carettochelyids?

4. l.216: the "crownward stem-lineage" term looks akward -> could you rephrase it?

5. l. 258: orbital ??? you probably mean orbit. I am not sure that sulcus would be a good term to describe a space.

6. l.255: Thus, we use terms from -> I suggest "we extend", and you can drop "although we note that it is incipient with respect to the pleurodire morphology".

7. l.260. You say that there is another similarity with pleurodires, but a few lines after you write "also an eclectic group of cryptodires" -> this similarity is therefore more likely to be not only shared with pleurodires, can you list cryptodira you are refering to (or provides a set of examples)?

8. l. 338: do you suggest it is destroyed or that it never existed ? maybe you can send the reader to the vomer part here "(see vomer section)".

9. l. 480: Well, but does that means that it had the same function than the processus TROCHLEARIS pterygoidei? Not sure that there is any functional homology here. It seems no from what you write juste after.

10. you are using "parabasisphenoid" and usually refer to this complex for the contact and comparison, I suggest to use basisphenoid or parasphenoid whenever you can since these two bones can be distinguished in that specimen.

11. l. 736-738 and next: does that change anything to select one or the other name ? does that mean that when the internal carotid artery enter anteriorly, we should stop homologize the opening with the foramen posterius canalis carotici interni ? or does that means that you want to homologize foramen posterius canalis carotici cerebralis with the canalis carotici interni as a whole ???? I think that the fpccc should be indicated on fig 7C. I have also a problem with fig 7E: how can we see faccc in this view ? it should be concealed by the lateral flange of the rbs (cf fig 7D)? You do not discuss the position of faccc in the text, I think it should be done. why not finally calling fpccc -> foramen ventralis canalis carotici cerebralis; and faccc : foramen dorsalis canalis carotici cerebralis ? Does the CT reveal the trajectory of the canal issuing from what you call faccc ?

12. l. 850...etc: it would be good to have a table summarizing similarities and differences among groups of paracryptodira that would be a good complement to the text.

13. l. 909: contact between Pterygoids and basioccipital : it was noted already in Gaffney 1979 for Glyptops; and Gaffney also noted this contact in Dorsetochelys (fig 28). Subsequent authors probably forgot about it, but it was already at least supposed that this contact was present in several mesozoic groups (not only in Baenids).

This is a signed review:
Julien CLAUDE
Institut des Sciences de l'Evolution de Montpellier.

---

## Round 0.2 · accepted · Accept

Great work, congratulations!

·

Basic reporting

The authors answered to all my comments satisfactorily and, specifically, they revised the segmentation of the maxilla/premaxilla bones. I am satisfied with the revised manuscript and, as such, I recommend it for publication as it is.

Experimental design

No comments.

Validity of the findings

No comments.